# The Chinese mitten crab genome provides insights into adaptive plasticity and developmental regulation

Zhaoxia Cui [1,2,3,12,13✉], Yuan Liu[2,3,4,12], Jianbo Yuan[2,3,4,12], Xiaojun Zhang [2,3,4,12], Tomer Ventura [5,12], Ka Yan Ma[6], Shuai Sun [7], Chengwen Song[2], Dongliang Zhan[8], Yanan Yang[1], Hourong Liu[2], Guangyi Fan[7], Qingle Cai[8], Jing Du[2], Jing Qin[6,9], Chengcheng Shi[7], Shijie Hao[7], Quinn P. Fitzgibbon[10], Gregory G. Smith[10], Jianhai Xiang [2,3,4], Tin-Yam Chan[11], Min Hui [2,3,4], Chenchang Bao[1], Fuhua Li [2,3,4,13✉] & Ka Hou Chu [6,13✉]

The infraorder Brachyura (true or short-tailed crabs) represents a successful group of marine invertebrates yet with limited genomic resources. Here we report a chromosome-anchored reference genome and transcriptomes of the Chinese mitten crab *Eriocheir sinensis*, a catadromous crab and invasive species with wide environmental tolerance, strong osmoregulatory capacity and high fertility. We show the expansion of specific gene families in the crab, including F-ATPase, which enhances our knowledge on the adaptive plasticity of this successful invasive species. Our analysis of spatio-temporal transcriptomes and the genome of *E. sinensis* and other decapods shows that brachyurization development is associated with down-regulation of Hox genes at the megalopa stage when tail shortening occurs. A better understanding of the molecular mechanism regulating sexual development is achieved by integrated analysis of multiple omics. These genomic resources significantly expand the gene repertoire of Brachyura, and provide insights into the biology of this group, and Crustacea in general.

[1] School of Marine Sciences, Ningbo University, Ningbo, China. [2] Key Laboratory of Experimental Marine Biology, Institute of Oceanology, Chinese Academy of Sciences, Qingdao, China. [3] Laboratory for Marine Biology and Biotechnology, Qingdao National Laboratory for Marine Science and Technology, Qingdao, China. [4] Center for Ocean Mega-Science, Chinese Academy of Sciences, Qingdao, China. [5] School of Science and Engineering, University of the Sunshine Coast, Sippy Downs, QLD, Australia. [6] Simon F. S. Li Marine Science Laboratory, School of Life Sciences, The Chinese University of Hong Kong, Shatin, Hong Kong, China. [7] BGI-Qingdao, BGI-Shenzhen, Qingdao, China. [8] 1GENE, Hangzhou, China. [9] School of Pharmaceutical Sciences (Shenzhen), Sun Yat-sen University, Guangzhou, China. [10] Institute for Marine and Antarctic Studies, University of Tasmania, Hobart, TAS, Australia. [11] Institute of Marine Biology and Center of Excellence for the Oceans, National Taiwan Ocean University, Keelung, Taiwan. [12] These authors contributed equally: Zhaoxia Cui, Yuan Liu, Jianbo Yuan, Xiaojun Zhang, Tomer Ventura [13] These authors jointly supervised this work: Zhaoxia Cui, Fuhua Li, Ka Hou Chu ✉email: cuizhaoxia@nbu.edu.cn; fhli@qdio.ac.cn; kahouchu@cuhk.edu.hk

rue or short-tailed crabs (infraorder Brachyura), with 7200
+ species described[1], constitute the largest group of dec-
apod crustaceans, one of the most successful animal taxa
worldwide. Brachyuran crabs occupy diverse niches from deep
oceanic to intertidal, freshwater, and terrestrial environments.
Crab morphology is characterized by developmental brachyur-
ization in which the tail (abdomen) at the larval stage becomes
reduced and folded beneath the adult cephalothorax, when the
individual develops from the final larval stage (megalopa) into a
juvenile crab[2]. This metamorphosis enables crabs to evolve body
plans dramatically different from those of other crustaceans. At
present, the molecular mechanism of brachyurization has not yet
been satisfactorily elucidated.

The Chinese mitten crab *Eriocheir sinensis* (H. Milne Edwards,
1853) has attracted considerable research attention due to its
importance in fisheries and aquaculture in China, as well as its
detrimental impact as an invasive species[3,4]. With its native range
in East Asia from Korea to South China, *E. sinensis* has invaded
European and North American waters, leading to significant
ecological and economic impacts[5,6]. *E. sinensis* has high fertility,
dispersal ability, and wide environmental tolerance, facilitating its
successful invasion[4]. These traits have also facilitated the rapid
expansion of its aquaculture across China, making it the most
important cultured crab globally, with net worth more than US
$10 billion per year[7]. A reference genome and extensive tran-
scriptome sequencing can provide a way to reveal the adaptive
plasticity of *E. sinensis*, i.e., its flexibility in terms of the capability
to cope with environmental changes[8,9].

An important trait underlying the success of *E. sinensis* is its
strong osmoregulatory capacity[10–12]. As a catadromous species,
the crab spends most of its life in freshwater, while mature adults

migrate to the sea for mating. After spawning and larval devel-
opment in coastal waters, their offspring migrate upstream back
to freshwater as juveniles (Fig. 1a)[13,14]. However, in *E. sinensis*,
evidence linking development with osmoregulation is lacking,
and the crucial genes related to osmoregulation have yet to be
identified.

Male sexual differentiation and maintenance of *Eriocheir
sinensis* are controlled by the androgenic gland (AG), an endo-
crine organ unique to the males of malacostracan crustaceans. AG
produces and secretes insulin-like androgenic gland hormone
(IAG) that regulates masculinity, which is considered the single
conserved sexual differentiating factor across Malacostraca[15]. As
monosex aquaculture of many crustaceans, such as all-female *E.
sinensis* culture, is preferred, a better understanding of the sexual
development pathway, especially the mode of action of IAG, is
imperative. At present, genetic sex markers are currently not
available for *E. sinensis* and it is not clear if sex change is
achievable through IAG manipulation. Multiomics analysis would
enhance our understanding on the molecular mechanisms of
sexual development of this ecologically and commercially
important species.

Here, we report a chromosome-anchored reference genome of
*E. sinensis*, representing a major addition to the decapod high-
quality genome assemblies, which are only available for the
Pacific white shrimp *Litopenaeus vannamei*[16], the marbled
crayfish *Procambarus virginalis*[17] and the swimming crab *Por-
tunus trituberculatus*[18]. While two draft genomes of *E. sinensis*
have been reported[19,20], little efforts have been made to relate the
genome assembly to the biology of the species. In this work, based
on comparative genomic analysis with other arthropods, the
highly adaptive plasticity of *E. sinensis* is linked with the

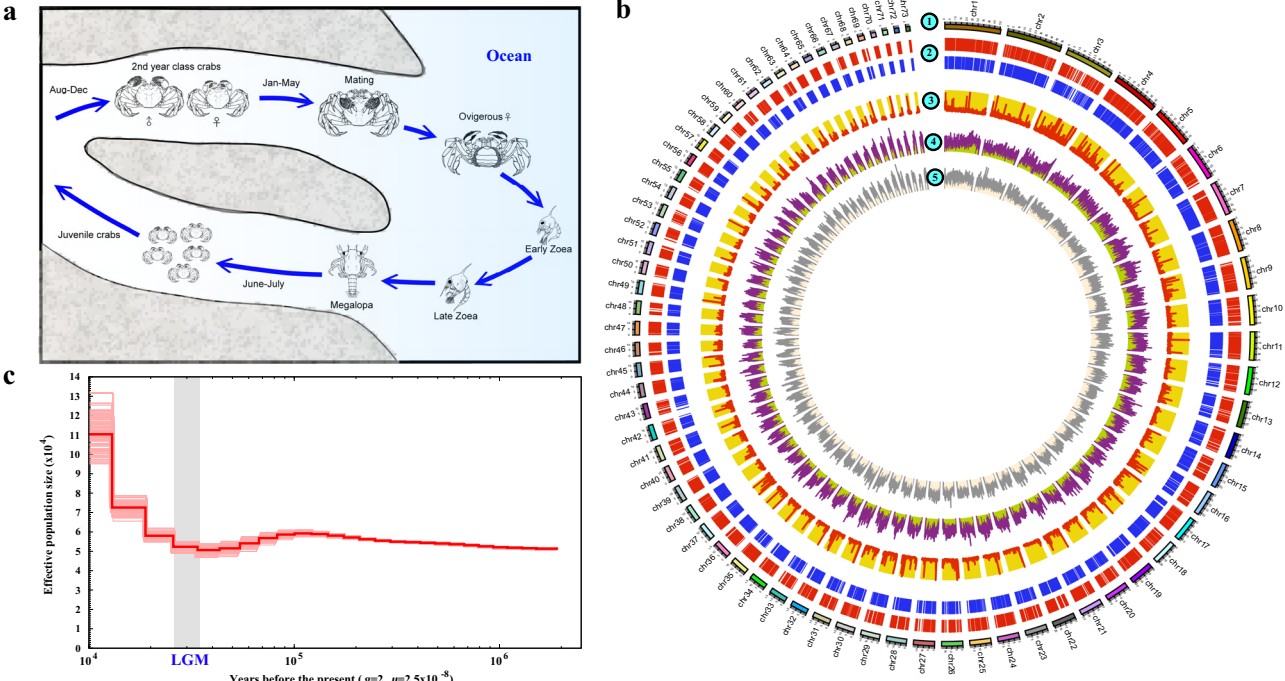

**Fig. 1 Characterization of the *Eriocheir sinensis* genome. a** The life cycle of *E. sinensis*. Adult crabs attain sexual maturity in estuaries followed by egg
hatching and then upstream migration of megalopae to rivers and lakes. **b** The genome landscape of *E. sinensis*. From outer to inner circles: b1, marker
distribution on 73 linkage groups at Mb scale; b2, protein-coding genes located on anchored scaffolds; b3, distribution of high- (>60%) and low-GC
(<20%) contents in 200-bp regions across the genome within a 100-kb sliding window; b4, microsatellite distribution (1–6 bp) across the genome drawn in
100-kb sliding windows; b5, SNP density of resequenced individuals drawn in 100-kb sliding windows within a 50-kb step, with polymorphism hotspot
regions (*P* < 0.05) colored in red. The genomic visualization was created using the program Circos. **c** Demographic histories of mitten crab reconstructed
using PSMC model. The period of the last glacial maximum (LGM, ~18,000–22,000 years ago) is shown as a gray bar. *g* (generation time) = 2 years; *μ*
(neutral mutation rate per generation) = 2.5 × 10$^{-8}$.

expansion of multiple stress-related gene families and F-ATPase family, which exhibits an expression pattern during development that documents its crucial role in osmoregulation. Examining this extensive data trove provides pioneering mechanistic insights into the developmental regulation of brachyurization in true crabs. Moreover, integrated analysis of multiple omic resources has elucidated the regulatory mechanism of IAG secretion and its signal transduction pathway.

## Results

**Genome sequencing, assembly, and characterization.** The main challenges of genome sequencing and assembly in marine invertebrates are high levels of heterozygosity and repetitive elements[21–23]. The *E. sinensis* genome contains 73 pairs of chromosomes ($2n = 146$)[24], which represent the second largest number of chromosomes in the reported arthropod genomes after the marbled crayfish. To yield a high quality reference genome, a male crab from six successive generations of inbreeding was used for sequencing by combining Illumina shotgun, PacBio SMRT, 10X Genomics Chromium sequencing and Hi-C technology. A 258× coverage of Illumina clean sequences and 35× coverage of PacBio long reads were used for hybrid assembly, and the scaffolding was accomplished by incorporating 106× coverage of 10X Genomics Chromium data (Supplementary Table 1 and Supplementary Fig. 1). To assemble the genome into chromosome level, ~132 Gb Hi-C data were used and 10,791 scaffolds were anchored to 73 chromosomes (Fig. 1b), accounting for ~80.82% of the genome assembly (~1.57 Gb) and nearing the size estimated by *k*-mer analysis (~1.45 Gb) and flow cytometry (~1.77 Gb, Supplementary Figs. 2 and 3). The assembly with a scaffold N50 size of 17.13 Mb was comparable to, or better than, those of other crustaceans (Supplementary Tables 2 and 3)[16,17,25,26].

The quality and integrity of the assembly were demonstrated by the mapping of over 93% of paired-end reads, 94% of PCR-amplified contigs, 95.4% of transcriptomic data, and 91.2% of 303 eukaryote genes by BUSCO-based completeness assessment (Supplementary Tables 4–7). Using our published high-density genetic linkage map[24] to validate the accuracy of the assembly of the chromosomes, we found that most of the linkage groups (67/73) were consistent with the assembled chromosomes (Supplementary Fig. 4). In particular, chr13 contained the deduced sex chromosome LG60 in the linkage map[24] with more sex-related regions, further suggesting a high quality of the chromosome assembly using Hi-C (Supplementary Fig. 5). This assembly represents a significant improvement on the published *E. sinensis* genome (Supplementary Figs. 6 and 7).

The *E. sinensis* genome encoded 28,033 protein-coding genes (Supplementary Fig. 8 and Supplementary Table 8), of which 93.17% were annotated based on known genes/proteins in the public databases (Supplementary Table 9). Repetitive sequences accounted for 45.30% of the assembly and were dominated by transposable elements (TEs) (566.69 Mb, 36.27%). As powerful drivers of genome evolution[27,28], the TEs in *E. sinensis* showed lower divergence than those in the amphipod *Parhyale hawaiensis* and the branchiopod *Daphnia pulex* (Supplementary Fig. 9), possibly as a result of relatively recent expansions. Microsatellites, the tandem repeats of importance to generate genetic variation underlying adaptive evolution[29], made up 6.92% of the *E. sinensis* genome (Fig. 1b and Supplementary Tables 10 and 11), a higher proportion than the values reported for other arthropods[30], with the single exception of the shrimp *L. vannamei* (Supplementary Fig. 10). GC-rich regions (GC content>60%) accounted for 4.68% of the 200-bp windows in the *E. sinensis* genome, higher than the proportions recorded in other reported decapod genomes. Further, they were approximately 4-fold and 47-fold higher

(respectively) than those of the water flea *D. pulex* and the scallop *Patinopecten yessoensis* (Supplementary Table 12). The high frequency of the GC-rich regions may be a feature of decapod genomes, as it was significantly higher than those in the genomes of 22 representative species. Compared with five other crustacean genomes, more genes were located in the GC-rich regions in *E. sinensis* (Supplementary Fig. 11).

Re-sequencing of 15 individuals from different locations in China provided a genome-wide scan of single nucleotide polymorphism (SNP) and short insertion/deletion (indel) polymorphism rate of 1.49% (Supplementary Tables 13–15). It is higher than that in the oyster *Crassostrea gigas*[22] (1.30%) and 10-fold higher than that in humans (0.14%)[31], indicating the high level of polymorphism and complexity of the *E. sinensis* genome. Demographic history analysis showed that *E. sinensis* maintained a relatively stable population size, followed by an obvious expansion which started after the last glacial maximum (LGM, ~20,000 years ago) (Fig. 1c). This may be related to the dramatic increase in area of the East China Sea-Yellow Sea following the rise in sea level.

**Gene family expansion and adaptive evolution in *E. sinensis*.** To analyze the diversification of crustaceans, we compared the *E. sinensis* genome with 10 other crustaceans, three insects and one chelicerate. Phylogenetic analysis based on 16 single-copy orthologous genes suggests *E. sinensis* (representing Thoracotremata) diverged from *Portunus trituberculatus* (representing Heterotremata) ~147.5 million years ago (Mya), and the pleocyematans diverged from dendrobranchiates (represented by *Litopenaeus vannamei*) ~280.6 Mya (Fig. 2a, Supplementary Table 16, and Supplementary Data 1). Comparative genomic analyses detected 27,836 families of homologous genes in Arthropoda, whereas species-specific genes made up a large proportion (Supplementary Fig. 12), possibly due to the few number of species under comparison in this largest, highly diverse animal phylum. A core set of 1389 gene families were shared by *E. sinensis* and three other decapod species (Fig. 2b), many of which were enriched in cell redox homeostasis, immune response and nucleotide metabolism (Supplementary Data 2).

As changes in gene copy number might support adaptive evolution, we examined the expansion of gene families in *E. sinensis* genome to explore potential mechanisms underlying the crab's adaptability. Compared with other arthropods, we identified 991 *E. sinensis*-specific and 2955 expanded gene families in this species (Fig. 2a). These gene families were predominantly involved in heat shock protein binding, oxidation-reduction process and various transporters (Supplementary Data 3–6), which might contribute to the crab's ability to overcome diverse stresses. Specifically, we found significant expansion of heat shock protein 70 (HSP70) (Fig. 2c), which is chaperone generally responsible for preventing damage to proteins in different stressful conditions. The *E. sinensis* genome was also enriched with gene families such as thioredoxin-like proteins (TXN) and cytosolic manganese superoxide dismutase (MnSOD) (Supplementary Table 17) that are important in modulating oxidative stress, which could result from the presence of xenobiotic, microbe-induced immune responses, and radiation[32]. Besides, ABC transporter, which was one of the families significantly expanded in the *E. sinensis* genome (Supplementary Table 17), plays a role in detoxification metabolism[33]. As an euryhaline crab and an invasive species that is able to withstand salinity fluctuation[34], desiccation during overland dispersal[35] and contaminants in polluted waters[36], the Chinese mitten crab's extraordinary ability to overcome these

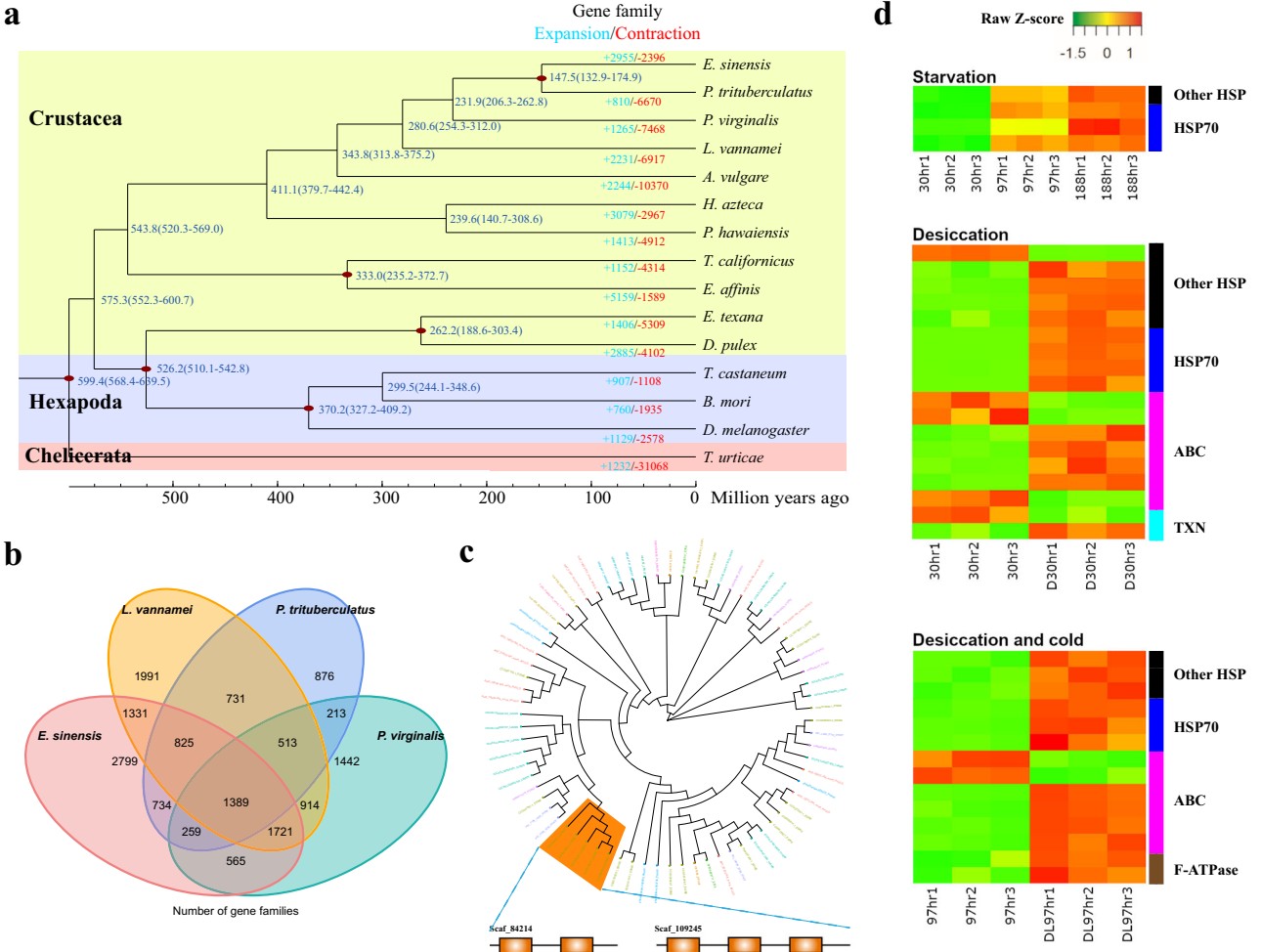

**Fig. 2 Comparative and evolutionary analyses of the *Eriocheir sinensis* genome. a** Gene family expansion/contraction analysis of 15 selected species. A total of 2955 gene families are expanded in *E. sinensis* relative to other arthropods. Expanded and contracted gene families are highlighted in blue and red, respectively. **b** Venn diagram of shared and unique gene families in four decapod species. **c** Expansion of HSP70 in the *E. sinensis* genome, which are also tandemly duplicated. **d** Expression patterns of HSP, ABC, F-ATPase, and TXN in response to starvation, desiccation and low temperature. 30hr, 97hr and 188hr, starvation for 30, 97, and 188 h at 20 °C; D30hr, desiccation at 20 °C for 30 h; DL97hr, desiccation at low temperature (5 °C) for 97 h.

stresses could be, at least in part, supported through the expansion of these gene families.

To further explore how these gene families contribute to stress responses in *E. sinensis*, we examined the changes in RNA expression in adult crabs subjected to starvation for 30, 97, and 188 h at 20 °C, desiccation at 20 °C for 30 h, and desiccation at low temperature (5 °C) for 97 h. The last stress condition, though unlikely in natural settings, is of interest because the mitten crab is well-known for its ability to withstand long term desiccation at low temperature during marketing. The sampling time for the experiment was determined based on time of first mortality (30 and 97 h) and LT50 (188 h) in pilot tests under desiccation. Results showed that stress resulted in significant ($q$-val < 0.05) elevation of HSP70 and other HSP genes, while significant changes in expression of TXN and ABC transporters were also observed though the patterns were not consistent across genes or conditions (Fig. 2d).

Therefore, the aforementioned species-specific and expanded gene families could be important to the *E. sinensis* lineage-specific environmental adaptations, enabling the crab to occupy diverse niches in a myriad of stressful conditions. In addition, the significant expansion of the F-ATPase gene family (Supplementary Table 17), which is known to be the prime producer of ATP in mitochondria[37], could enhance the capacity of *E. sinensis* to

support various energy demanding processes such as stress response, locomotion, and osmoregulation (see next section).

**Catadromous migration of *E. sinensis* with expansion of F-ATPase family.** *E. sinensis* is a catadromous species which migrates from freshwater to the sea to spawn and complete its larval development. Upon completion of the larval stages, the offspring then migrate back to freshwater habitats as juveniles. Adult sexual maturity and juvenile development are hindered when the crabs are maintained at a constant salinity, while an increase of salinity during the post-metamorphic freshwater stage can promote sexual maturity (Fig. 1a)[14]. Salinity variation therefore plays a key role in the development and reproduction of *E. sinensis*.

The ability of *E. sinensis* to engage in catadromous migration is attributed to its well-developed osmoregulatory capacity. Copy numbers of different osmoregulation-related genes were similar across distantly related arthropods inhabiting various terrestrial and aquatic habitats, with the exception of F-type H⁺-ATPase (F-ATPase). This was significantly expanded in *E. sinensis* (102 members), showing more than a fivefold increase over its occurrence in other marine crustaceans (Fig. 3a). For reference, even in the marine crab *P. trituberculatus*, the copy number of

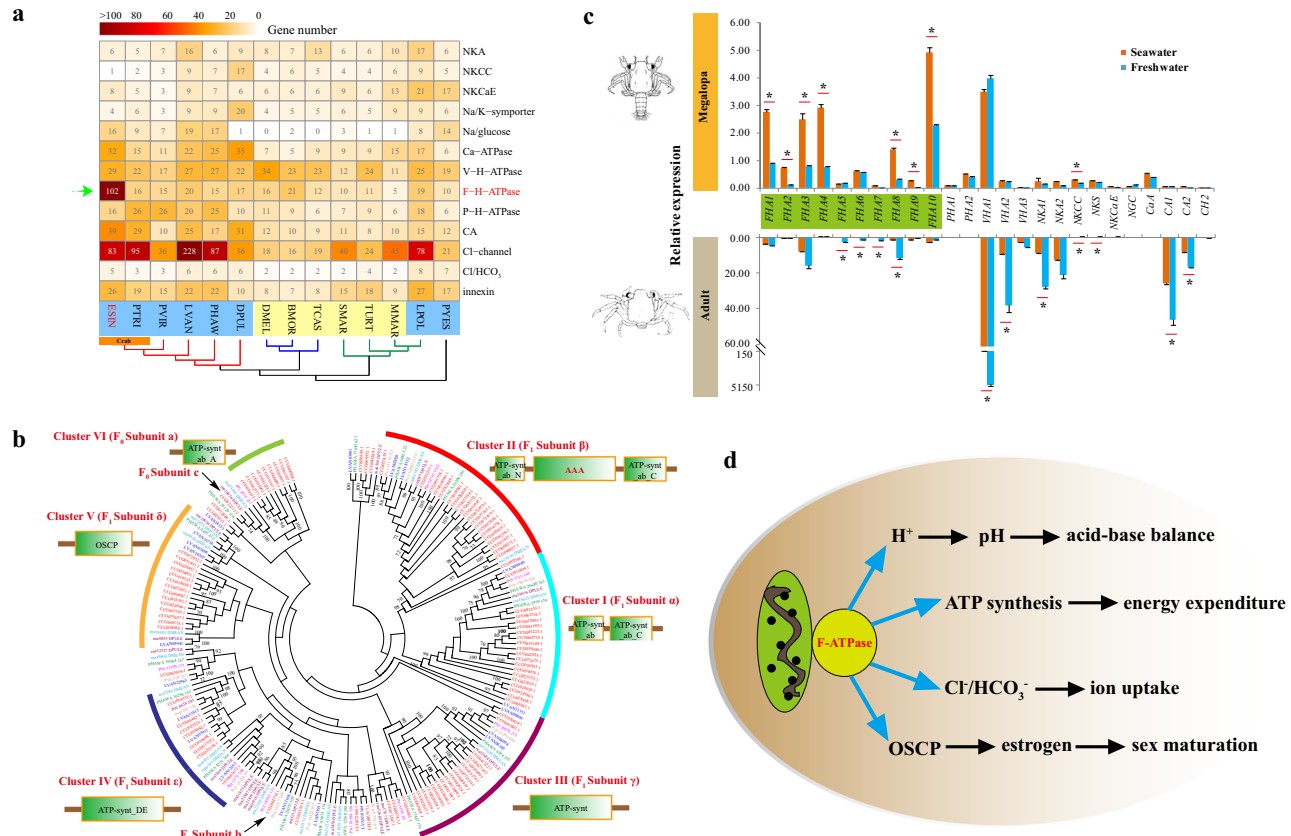

**Fig. 3 Osmoregulation related genes in the *E. sinensis* genome. a** Gene copy numbers of different osmoregulation related genes among arthropods[123, 124]. F-H-ATPase is the only gene family that significantly expands in the *E. sinensis* genome. The species in the comparison includes *E. sinensis* (ESIN), *Portunus trituberculatus* (PTRI), *Procambarus virginalis* (PVIR), *Parhyale hawaiensis* (PHAW), *Daphnia pulex* (DPUL), *Drosophila melanogaster* (DMEL), *Bombyx mori* (BMOR), *Tribolium castaneum* (TCAS), *Strigamia maritima* (SMAR), *Mesobuthus martensii* (MMAR), *Tetranychus urticae* (TURT), *Limulus polyphemus* (LPOL), and *Patinopecten yessoensis* (PYES). The osmoregulation related genes include Na$^+$/K$^+$ ATPase (NKA), Na$^+$/K$^+$/2Cl$^-$ cotransporter (NKCC), Na$^+$/K$^+$/Ca$^{2+}$ exchanger (NKCaE), Na$^+$/K$^+$ symporter (Na/K-sympoter), sodium/glucose cotransporter (Na/glucose), Ca$^{2+}$ transporting ATPase (Ca-ATPase), Vacuolar H$^+$-ATPase (V-H-ATPase), F-type H$^+$-transporting ATPase (F-H-ATPase), P-type H$^+$-transporting ATPase (P-H-ATPase), Carbonic anhydrase (CA), chloride channel protein (Cl-channel), Na$^+$-independent Cl/HCO$_3$ exchanger (Cl/HCO$_3$-exchanger), and innexin. **b** Expansion of F-ATPase in the *E. sinensis* genome. F-ATPase consists of two domains: F$_O$ domain (containing three subunits: **a**–**c**) that integrates in the membrane to promote passive H$^+$ transport, and F$_1$ domain (containing five subunits: α, β, δ, γ, ε) which is easily released from the membrane, catalyzing hydrolysis of ATP[44]. The expanded genes in each cluster share similar gene structures. The domains of each cluster were predicted by SMART. Genes from different species are represented by different colors: *E. sinensis* (red), *P. hawaiensis* (green), *Daphnia pulex* (orange), *Drosophila melanogaster* (blue). **c** The expression of osmoregulation related genes in megalopae (upper) and adults (lower) based on qPCR. Asterisks indicate the significant differences in gene expression between crabs in seawater (orange) and freshwater (blue). FHA F-H-ATPase, PHA P-H-ATPase, VHA V-H-ATPase, NKS Na/K-sympoter, CH Cl/HCO$_3$-exchanger. **d** Potential functions of F-ATPase in osmoregulation and development of *E. sinensis*. F-ATPase performs its function through regulating acid-base balance, energy expenditure, ion uptake, and estrogen-targeting in sex maturation.

F-ATPase (16 members) was much lower than that of *E. sinensis*. F-ATPase is known to be the prime producer of ATP, using the proton gradient generated by oxidative phosphorylation in the mitochondria[37]. Together with V-type and P-type ATPase, F-ATPase is one of the three major classes of proton pumps in eukaryotic cells[38,39]. V-ATPase is an important osmoregulatory gene in *E. sinensis*, along with Na$^+$/K$^+$-ATPase (NKA), well known as the Na$^+$/K$^+$ pump[40,41]. As with NKA and V-ATPase, the protein content and the activities of F-ATPase are identified in the posterior gills (involved in ion absorption) rather than in their anterior counterparts (not involved in ion absorption), and the activities of these three ATPases are higher in diadromous crabs than in freshwater crabs[42], indicating their role in osmoregulation.

Ka/Ks analysis of the orthologs between the two crabs (*E. sinensis* and *P. trituberculatus*) indicated only six osmoregulation-related genes were under positive selection. None of the genes

encoding NKA, V-type and P-type ATPases were positively selected, but two genes encoding F-ATPase have ω > 1 (Supplementary Data 7). Furthermore, the ratio of activities[43] between F-ATPase (41.2) of the posterior and anterior gills is more than five times higher than those of NKA (7.5) and V-ATPase (6.2), suggesting that F-ATPase may be positively selected and play a more significant role than the two other ATPases in the osmoregulatory functioning of the *E. sinensis* gill. F-ATPase is a large complex composed of several subunits, which were equally rather than specifically expanded in the *E. sinensis* genome (Fig. 3b), suggesting that they were of equal functionality, such that the gene dosage of F-ATPases was equally increased. These subunits also displayed similar expression patterns across different developmental stages and tissues ($p < 0.0001$) (Supplementary Fig. 13), far more consistent than other osmoregulation-related genes (Supplementary Figs. 14–16). The *b* and *c* subunits of F0-ATPase complex were the only two single-copy genes that

have not been expanded (Fig. 3b). However, the F0-ATPase subunit $c$ was found to be under positive selection ($\omega = 2.4039$) (Supplementary Data 7), and had higher expression level (FHA10 in *E. sinensis*) than other F-ATPase genes (Fig. 3c). Taken together, the subunits co-expansion, co-expression during development, positive selection and high activity in the posterior gills, all point to F-ATPase being a gene family crucially related to osmoregulation in *E. sinensis*.

Unlike V-ATPase but similar to NKA, F-ATPase was not only upregulated during the salinity decrease after megalopa, but was also highly expressed at the zoea stages (Supplementary Figs. 13 and 16). F-ATPase was also specifically expressed in gills and muscles, indicating its important functions in both development and osmoregulation. To further identify its functions during larval development, we conducted qPCR analyses of selected osmoregulatory genes and F-ATPases (Supplementary Data 8) in megalopa and adult crabs acclimated to freshwater and seawater. As expected, NKA, V-ATPase and many other osmoregulation related genes were differentially expressed in adults, but not in megalopae (Fig. 3c), suggesting different osmoregulatory mechanisms between megalopae and adults. In comparison with other osmoregulation-related genes, F-ATPases were mostly (except for FHA5, FHA6, and FHA7) downregulated more significantly in megalopae than in adults, indicating their important osmoregulation roles during megalopa development. By contrast, FHA5, FHA6, and FHA7 were more significantly downregulated in adults than in megalopae, a pattern similar to that found in other osmoregulation-related genes. Thus, F-ATPase may not only play important osmoregulation roles in megalopae, but also have general functions of osmoregulation in adults. Besides, according to previous reports[37,44,45], the functions of F-ATPase related acid-base balance and saving energy expenditure may also important for the osmoregulation of crabs (Fig. 3d). In summary, the significant expansion of F-ATPase is attributed to the catadromous migration of *E. sinensis*.

**Split Hox cluster and the regulation in brachyurization metamorphosis.** Crustaceans present the most impressive diversity in body plan among Arthropoda[46]. Hox genes encode homeodomain-containing transcription factors (TFs) that play crucial roles in the anterior–posterior (AP) patterning of the bilaterian animal body[47]. The *E. sinensis* genome contained all ten canonical Hox genes (*lab, pb, Hox3, Dfd, Scr, ftz, Antp, Ubx, Abd-A*, and *Abd-B*) found in the arthropod ancestor[48] (Fig. 4a and Supplementary Fig. 17). Differing from the conventional compact Hox cluster in many genomes[49], the Hox genes of *E. sinensis* were found on four separate genomic scaffolds with flanking non-Hox genes in the genome (Supplementary Table 18). These four Hox gene-containing scaffolds were located on the same chromosome (chr21), spanning approximately 5.3 Mb (Supplementary Data 9), indicating that the crab has a loose Hox gene cluster, like many other arthropods[50]. Nevertheless, its spatial collinearity is conserved with most bilaterians. The expression of *E. sinensis* Hox genes also exhibited temporal collinearity during the crab's embryonic development (Fig. 4b). Unlike the whole-cluster temporal co-linearity (WTC) in vertebrates, the expression pattern of Hox genes in *E. sinensis* (Fig. 4b) is similar to the subcluster-level temporal collinearity (STC) in scallop, oyster, shrimp, and sea squirt[51]. The STC may be caused by the split Hox cluster, which is believed to be related to their complex metamorphosis in development[52–54].

Hox genes *Ubx, Abd-A*, and *Abd-B*, which are involved in specifying posterior thoracic and abdominal development[49], gradually increased in expression from the zoea stage to the postlarval stage in *L. vannamei*[55], and high expressions were also found in the eastern spiny lobster *Sagmariasus verreauxi* between the phyllosoma and juvenile stages (Fig. 4c). In *E. sinensis*, however, the expression of these three Hox genes decreased in the equivalent developmental stages, from the original zoea stage (Ozs) to the juvenile instar (J1) stage, correlating with the brachyurization metamorphic transition from late megalopa (LM) to J1 (Fig. 4c). In crustaceans, these posterior Hox genes are required for the formation of the abdomen and their limbs[56]. In Cirripedia, the loss of *Abd-A* has been correlated with the lack of an abdomen in this lineage[57]. Knockout of *Abd-A* by CRISPR/Cas9 system in the amphipod *P. hawaiensis* produces a simplified body plan characterized by a loss of abdominal appendages[58]. In *E. sinensis*, *Abd-A* expression was detected in the prolegs primordium on the LM and J1 stages (Supplementary Figs. 18 and 19). Since the main morphological changes are usually caused by the change of Hox gene expression pattern, we speculate that the abdominal degeneration of crabs should be associated with the low level of these Hox genes during brachyurization metamorphosis.

Four conserved arthropod miRNAs, i.e., *miR-993, miR-10, miR-iab-4*, and *miR-iab-8*, were identified in the Hox genes cluster of the *E. sinensis* and *P. trituberculatus* genomes (Fig. 4a). Interestingly, two copies of *miR-iab-4/8* were present in conserved synteny between *Abd-A* and *Abd-B* (Fig. 4a and Supplementary Table 18), making the two crabs the third and fourth reported arthropod species with duplicated copies of Hox-associated miRNAs[50,59]. In most insect species, *miR-iab-4* negatively regulates *Abd-A* and *Ubx*[60], while *miR-iab-8* regulates *Abd-A* and *Abd-B*[61], with ectopic expression of *miR-iab-4* and *miR-iab-8* inducing homeotic phenotypic transformations[62,63]. In *E. sinensis*, both *miR-iab-4* and *miR-iab-8* were up-regulated in the abdomen of LM and J1, while *Ubx, Abd-A*, and *Abd-B* were down-regulated in the abdomen of these stages (Fig. 4d), suggesting that these miRNAs can regulate the posterior Hox genes and facilitate the brachyurization metamorphosis.

Brachyurization involves dramatic morphological, physiological, and behavioral changes, orchestrated by complex gene networks, rather than by individual "morphology genes"[64]. In this study, we identified four Hth (Homothorax, a Hox cofactor) genes, whose transcripts originated from a single gene with different splicing patterns (Supplementary Fig. 20). Hox genes might therefore regulate various downstream genes through binding different Hth proteins, or other cofactors (Exd, Otx2, and Zfh1). These co-TF genes were slightly upregulated from the repression of posterior Hox genes, *Ubx, Abd-A*, and *Abd-B*. We observed that the expression of some segment-polarity genes, such as *Gsb, meis1*, and *twist*, also changed significantly during the brachyurization metamorphic transition (from LM to J1), while transcripts encoding downstream factors of the posterior Hox genes (*pax3, pou6f2, pitx, msx, col, ect*.), were down-regulated (Fig. 4e and Supplementary Data 10). During brachyurization metamorphosis (from LM to J1) of *E. sinensis*, many genes in the pathways of muscle contraction, oxidative phosphorylation, lipid metabolism, and calcium signaling displayed significant downregulation in the abdomen, while the PI3K-Akt and DNA replication pathway genes showed upregulation (Supplementary Fig. 21). The expression of many muscle-related genes and energy metabolism-related genes decreased, while that of the apoptosis-related and neurodegenerative-related genes increased (Supplementary Data 11). Ultimately, these regulations were reflected in the genetic function for brachyurization, abdominal muscle degeneration, abdominal nervous system specification, limb repression, and gonad cell lineage control (Fig. 4e).

It is noteworthy that crab-like body form (i.e., abdomen reduced and bent beneath cephalophorax) is not unique in Brachyura, but has independently evolved at least three times within Anomura (hermit crabs and their allies)[65–72]. The repeated convergent evolution of crab-like body form in

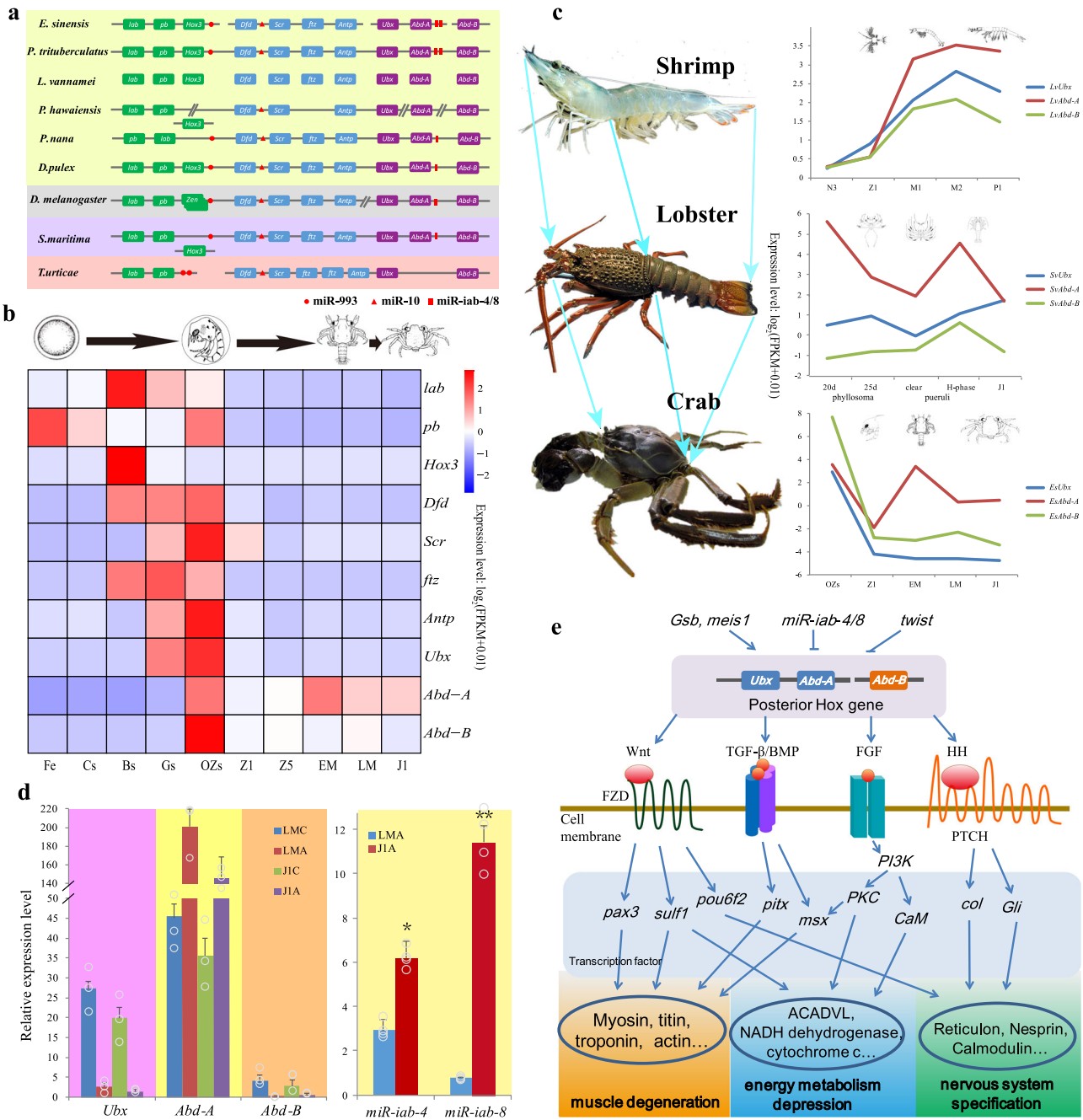

**Fig. 4 The conserved homeobox gene clusters and their expression and regulation in *E. sinensis*. a** Genomic organization of Hox genes in *E. sinensis* and selected arthropods. Different colored boxes indicate different Hox gene groups, including anterior (green), central (blue), and posterior (purple). Red shapes indicate the position of miR-993, miR-10 and miR-iab-4/8 genes. Split lines indicate that the cluster is located on different scaffolds. Oblique lines indicate the regions of the Hox cluster that are non-contiguous or interrupted on the same scaffold. **b** Temporal expression of *E. sinensis* Hox cluster genes. Fe fertilized egg, Cs 2–4 cell stage, Bs blastula stage, Gs gastrula stage, Ozs original zoea stage, Z1 zoea I, Z5 zoea V, EM early megalopa stage, LM late megalopa stage, J1 the first juvenile instar. **c** The expression and comparison of *Ubx*, *Abd-A*, and *Abd-B* genes in shrimp *L. vannamei*, lobster *S. verreauxi* and crab *E. sinensis*. The photo of *S. verreauxi* was provided by John Booth, National Institute of Water and Atmospheric Research, New Zealand. **d** The expression of posterior Hox genes (*Ubx*, *Abd-A*, and *Abd-B*) and miRNAs (*miR-iab-4* and *miR-iab-8*) in the cephalothorax and abdomen of the late megalopa (LM) and juvenile instar (J1) stages. LMC represents the cephalothorax of LM, and LMA represents the abdomen of LM, J1C represents the cephalothorax of J1, J1A represents the abdomen of J1. Error bars represent the mean ± S.D. ($n = 3$ in Hox expression and $n = 4$ in miRNA expression). Significant differences across LMA (two-tailed Student's *t*-test) are indicated with an asterisk at $P < 0.05$, and two asterisks at $P < 0.01$. **e** The speculative regulation process of the posterior Hox gene during the brachyurization metamorphosis in *E. sinensis*.

Decapoda, a phenomenon coined "carcinization" by Borradaile[65], has been frequently studied since the second half of the nineteenth century, but has thus far been limited to morphological perspective[71]. Here, we provide the first model of molecular mechanisms underlying the brachyurization metamorphosis of Brachyura: the split Hox clusters, *miR-iab-4/8* duplication and segment-polarity genes in the genome might tighten the regulation of the posterior Hox genes, leading to the degeneration of the abdomen giving rise to the distinctive body configuration of true crabs. Although good quality genome or transcriptome of anomurans are not yet available for comparison, our model would nonetheless serve as a pivotal foundation for future research to examine if, and to what extent, carcinization in decapods has resulted from evolution involving similar molecular pathways and genomic features.

**Crustacean specific androgenic gland and secretion regulation**. Unlike vertebrates, male sexual differentiation and maintenance in malacostracan crustaceans is fundamentally controlled by the androgenic gland (AG), a unique male crustacean endocrine organ separated from the gametogenic organ (testis)[73–75]. The AG is known to secrete the insulin-like androgenic gland hormone (IAG), known to induce masculinization and maintain male characteristics[15]. This endocrine regulation of sexual differentiation in crustaceans is unique among the animal kingdom, and yet many detailed processes and mechanisms remain obscure.

In *E. sinensis*, the AG was located bilaterally on the surface of the ejaculatory ducts (ED) posterior portion (Supplementary Fig. 22). To search for the IAG receptors and the TFs that trigger IAG transcription, we constructed a gene co-expression network from 29 transcriptome datasets and identified midnightblue module as the only AG-related module (Supplementary Figs. 23 and 24, and Supplementary Data 12). Several neurotransmitter-regulated genes, such as *Dop2Rs* (dopamine D2-like receptor), were members of this module, with specific expression in the AG (Supplementary Fig. 25 and Supplementary Data 13). Of these, *KCNN2*, *KCNN3*, *GRIN2B*, and *CADN* regulating neurotransmission and neurotransmitter release, were among the top-ranked hub genes (Fig. 5a and Supplementary Data 12 and 13), suggesting neural-related genes are key regulators of AG function in *E. sinensis*. More interestingly, many predicted target genes of miRNAs differentially expressed between the AG at synthesis (SY) and secretion (SE) phases are key molecules involved in neurogenesis, axon guidance and dendrite morphogenesis, identified by an integrated analysis of miRNA and mRNA expression profiles in SY and SE (Supplementary Figs. 26 and 27, and Supplementary Data 14). These results reflected the role of miRNAs in regulating the innervation of AG and further supported a closer connection between neurotransmission and AG function.

Analysis of the midnightblue module showed *IAG*, *iDMY* (Y-linked iDmrt1 paralogue), *InR1* and *InR2* (insulin-like receptors) were the most important hub genes with the highest intramodular connectivity (Fig. 5a and Supplementary Data 13), suggesting they are key regulators of AG development and function in *E. sinensis*. IAG, a key masculine AG factor, had two copies in the *E. sinensis* genome with IAG1 involved in male sexual differentiation (Supplementary Figs. 28 and 29). iDMY, a TF homolog of the first sex-linked Dmrt found in invertebrates[76], potentially regulates *IAG* transcription, supported by the putative Dmrt binding sites located at the promoter sequence of *IAG*, similar expression pattern of *iDMY* and *IAG* in sex distinguished larvae, and reduced expression of *IAG* following *iDMY* knockdown (Supplementary Figs. 30–32). InRs, a subfamily of insulin and IGF receptors (receptor tyrosine kinases, RTKs), mediate the IAG signaling pathway, as confirmed by recent experiments[77–79].

Interestingly, IAGs clustered with members of the relaxin subfamily (Fig. 5b, Supplementary Fig. 33, and Supplementary Data 15) but unlike relaxins, which operate through G protein coupled receptors (GPCRs), IAGs operated via RTKs (Supplementary Fig. 34). Various studies have shown that Arg-3X-Arg-2X-Ile/Val motif in the relaxin B chain and conserved TyrA19 in the insulin A chain are crucial for receptor binding and biological activity[80,81]. Crustacean IAGs lacked the relaxin-specific motif but retain the key amino acid Tyr (Fig. 5c and Supplementary Fig. 35). In addition, crustacean IAGs had ValB17 and GluA5 at the same sites as in insulins and IGFs (Fig. 5c). These properties together suggest that IAGs are homologous to relaxins but bind to insulin and IGF receptors due to change of key amino acids.

In summary, the IAG synthesis was proposed to be induced by dopamine binding to Dop2Rs (dopamine D2-like receptor) and a GPCR Mth2 (methuselah) presented on the plasma membrane of AG cells (Fig. 5d), which are known as insulin release-related receptors in *Drosophila* and mammals[82,83]. As suggested in insulin secretion[84], the activated Mth2 and Dop2Rs might trigger or inhibit cAMP signaling pathway, thus modifying the phosphorylation of iDMY, Sox15, CREB, and PDX-1 through protein kinase A (PKA). The activated TFs then translocated into the nucleus and triggered the transcription of *IAG*. To maintain the extremely high expression of *IAG* in AG cells, secreted IAG might enhance its own transcription through a positive feedback loop, in which it might bind insulin-like receptors (InRs) on AG cells and activate a distal enhancer via signaling pathways (Fig. 5d). Interestingly, we found several well-known pathways enriched with differentially expressed genes and proteins, such as neuroactive ligand-receptor interaction, PI3K-Akt, MAPK, relaxin pathways, and cAMP signaling pathway during AG development (Supplementary Data 16 and 17). These could serve as the upstream pathway activating TFs of *IAG*. These DEGs were further confirmed by RNA-seq data of eyestalk ablated *E. sinensis* (Supplementary Table 19 and Supplementary Data 18). Eyestalk ablation induces hypertrophy of the AG, which can strengthen the IAG synthesis and secretion. Then, it was established that the IAG signal was passed through the cross-talk of cAMP, cGMP, and calcium signaling pathways. All these reactions led to the mass synthesis and secretion of the IAG protein from the AG.

## Discussion

There have been few genomes reported either for decapods in general or for crabs in particular, despite their ecological and economic importance. The Chinese mitten crab *E. sinensis* is of major interest, because it is the most important cultured crab and also it is one of the world's 100 worst invasive alien species[85]. In this study, we provide insights into its unique adaptive plasticity which could be supported by the marked expansion of gene families that are related to stress responses. We also detect significant expansion of F-ATPase, which could be an important osmoregulatory gene for adaptation to catadromous migration. Coupled with other $H^+$ pumps and osmoregulatory genes, F-ATPase may participate in ion uptake as well as energy production, and hence assist in the active hyper-osmoregulatoin during the migration. F-ATPase has also been well characterized as a pH regulator of cell environment[44]. Furthermore, it can cooperate with $Cl^-/HCO_3^-$ exchanger, which shows a similar expression pattern, to maintain the cytoplasmic acid-base balance, that might be particular pivotal during *E. sinensis*'s migration to (often stressful) habitats. The crab's ability to thrive in diverse hostile conditions (e.g., salinity fluctuations, desiccation) have helped it become a successful colonizer of new ecological niches. Due to the high invasive capability of *E. sinensis*, this information will be valuable in informing management strategies.

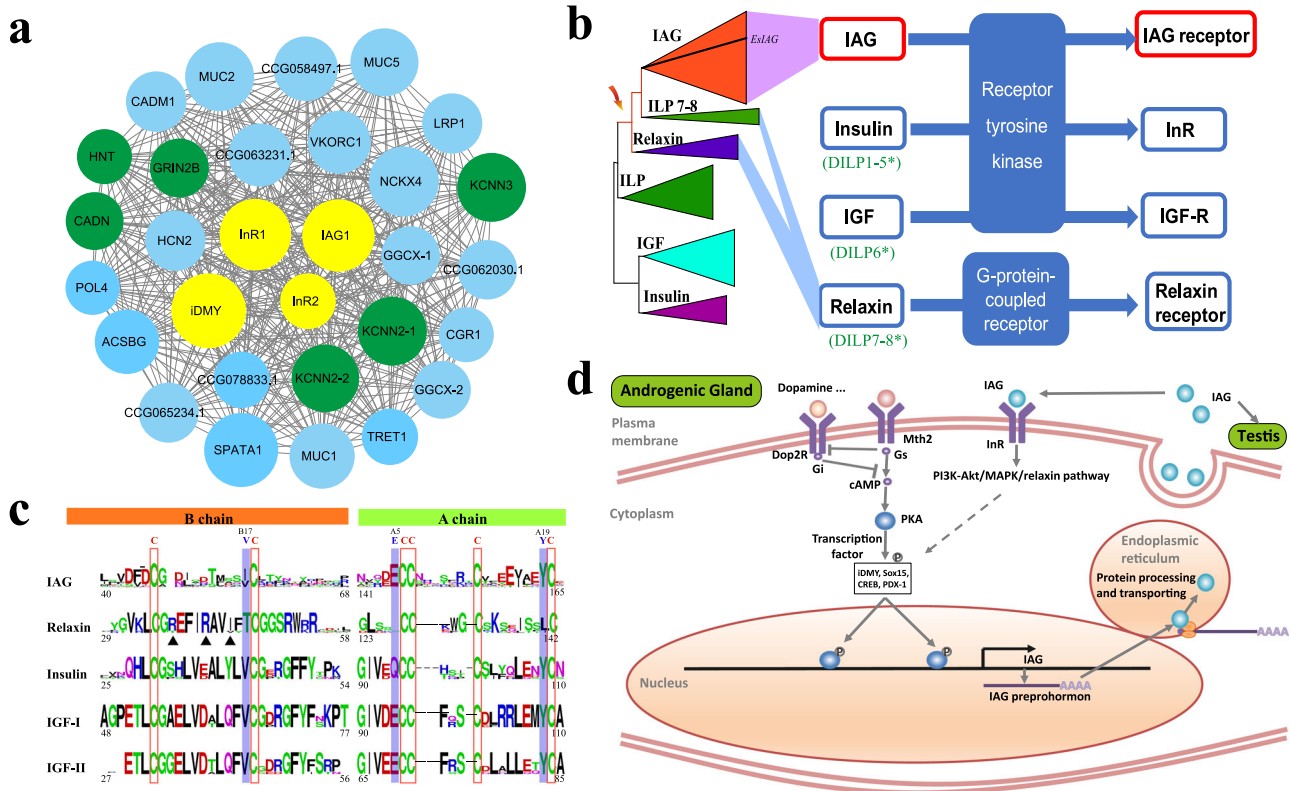

**Fig. 5 The molecular regulation of AG. a** The co-expression network of the AG-related module (midnightblue, see Supplementary Data 13). The midnightblue module contains 216 genes, of which 30 genes with the highest intramodular connectivity are chosen for network display. IAG-related receptors and transcription factors are labeled in yellow. The neuron or axon-related genes are labeled in green. **b** Insulin-like peptides and their receptors. The insulin-like peptide (ILP) is subdivided into IAG, insulins, insulin-like growth factors (IGFs) and relaxins on the basis of primary structure, processing and receptor binding preferences. **c** Conserved sites of crustacean IAG, and vertebrate relaxin, insulin and IGF-1 and IGF-II. The key functional sites for receptor binding are indicated by triangles. The amino acid numbers are with reference to IAG of *Litopenaeus vannamei* and other insulin members of human[125]. **d** The predicted synthesis and secretion pathway of IAG. Mth2 methuselah, PKA protein kinase A, Dop2R dopamine D2-like receptor, Gs guanine nucleotide-binding protein G(s) subunit alpha, Gi guanine nucleotide-binding protein G(i) subunit alpha, iDMY invertebrate sex-linked (Y-linked) Dmrt gene, Sox15 SRY-Box 15, CREB cyclic AMP-responsive element-binding protein, PDX-1 insulin promoter factor 1, InR insulin-like receptor.

From the evo-devo perspective, we provide the first evidence for the mechanism underlying developmental brachyurization. This metamorphosis enables brachyurans to evolve body plans dramatically different from those of other decapod crustaceans. The *E. sinensis* genome has all ten cannonical Hox genes in the arthropod ancestor, found in four separate scaffolds located on the same chromosome, indicating a loose Hox gene cluster. The expression of the Hox genes during development exhibits subcluster-level temporal collinearity, similar to other invertebrates. While the Hox genes *Ubx*, *Abd-A* and *Abd-B* that are involved in specifying posterior thoracic and abdominal development, increased in expression from the larval to postlarval or juvenile stages in shrimp and lobster, the expression of these three genes decrease from the larval to juvenile stage in *E. sinensis*. We also identify putative miRNAs and segment-polarity genes that regulate these Hox genes and facilitate brachyurization, as well as those genes on the downstream of the Hox genes that lead to tail degeneration during metamorphosis. To conclude, brachyuriation is mediated through down-regulation of the Hox genes at the megalopa stage.

During sexual development of decapods, the AG, a unique male crustacean endocrine organ, is known to induce and maintain masculinization through production and secretion of IAG[15,73,75]. We propose here a regulatory mechanism explaining the high IAG expression capacity of the AG and key components in the plausible IAG signal transduction pathway. Analysis of the

genes found in AG-related module suggests neural-related genes are key regulators of AG function in *E. sinensis*. This finding is further supported by the predicted target genes of miRNAs differentially expressed between the AG at synthesis and secretion phases, which play key roles in neurogenesis, axon guidance and dendrite morphogenesis, indicative of the key role miRNAs play in regulating the AG innervation. These results show a clear link between neurotransmission and AG function. In sum, our findings highlight the feedback loop of mass synthesis and secretion of IAG from AG, providing insights into the IAG regulatory mechanism of sexual differentiation in crustaceans.

In conclusion, the genome and transcriptomes of *Eriocheir sinensis* reported here give insights into brachyurization and sexual development of decapod crustaceans. Furthermore, as the crab is a successful invasive species, these data increase our understanding of adaptive plasticity underlying invasion biology.

## Methods

**Genome sequencing and assembly**. A male mitten crab *E. sinensis*, an inbred from six generations of small-size population mating produced by Panjin Guanghe Crab Industry Co., Ltd, was used for genome sequencing. All animal studies and procedures were approved by the Animal Ethics Committee [2020(37)] at Institute of Oceanology, Chinese Academy of Sciences (Qingdao, Shandong, China). High-quality genomic DNA was extracted from the muscle of the crab using Plant Genomic DNA Kit (TIANGEN, DP305) in accordance with the manufacturer's protocol. For Illumina sequencing, short-insert paired-end (PE) (250, 500, and 800 bp) and long mate-paired (MP) (2, 5, and 10 kb) DNA libraries were constructed in

accordance with the manufacturer's instructions (Illumina, San Diego, California, USA). Sequencing for the PE libraries was performed on the Illumina HiSeq4000, and for long MP libraries on the HiSeq2500. To obtain long reads for promoting genome assembly, Pacific Biosciences RS II (Pacific Biosciences, Menlo Park, California, USA) was used for sequencing. Five 10 kb SMRTbell libraries were prepared and sequenced using the C4 sequencing chemistry and P6 polymerase. A new assembly strategy HABOT[86] (Hybrid Assembly Based on TGS) (1gene, Hangzhou, Zhejiang, China) was used for hybrid assembly of high-fidelity short Illumina sequences and long PacBio reads. Scaffolding was accomplished by incorporating 10× Genomics Chromium data (see Supplementary Fig. 1 for details). A Hi-C library was constructed and sequenced by BGISEQ-500 (BGI, Qingdao, China) to link scaffolds to chromosomes with a 3d-dna[87] pipeline. Juicerbox[88] was used to modify the order and directions of some scaffolds in a Hi-C contact map and to help in the determination of chromosome boundaries. The genome size of *E. sinensis* was estimated using flow cytometry and *k*-mer analysis. The integrity of the final assembly was assessed using four data sets: contigs validated with PCR, transcriptome data, 274 complete *E. sinensis* coding DNA sequences (CDS) from NCBI, and a 1369-BUSCO metazoan subset of genes (303 from Eukaryota and 1066 from Arthropoda).

**Assembling chromosomes using Hi-C in comparison to linkage mapping and map integration**. We previously constructed a high-density linkage map of *E. sinensis* with 10,358 SNP markers using the 2b-RAD methodology[24]. To anchor scaffolds to chromosomes, marker sequences were aligned back to the genome assembly using BLAT v36[89] with the parameter -tileSize=7. Only markers with unique location and alignment length coverage >85% in the assembly were retained. In cases where scaffolds were in conflict with the genetic map, we manually checked them and broke the scaffolds at conflicts with low-coverage and gaps. Final anchoring and orientation of the scaffolds to corresponding linkage groups was conducted using ALLMAPS v0.6.9[90] with default parameters. Finally, we compared the order and orientation of the scaffolds between Hi-C assembled chromosomes and linkage map assembled chromosomes using in-house script.

**Repetitive sequence and GC bias analysis**. Both homology-based and de novo predictions were used to identify transposable elements (TEs) in the genome. For homology-based analysis, we used RepeatMasker v4.0.6 and RepeatProteinMask v4.0.6 (http://www.repeatmasker.org) to detect TEs in the Repbase library[91]. De novo TEs prediction was performed with RepeatModeller v1.0.8 (http://www.repeatmasker.org/RepeatModeler.html). Tandem repeats were identified using Tandem Repeats Finder v4.0.7 (http://tandem.bu.edu/trf/trf.html)[92]. To survey the base composition and distribution of the assembly genome, we determined the GC content in 200-base non-overlapping sliding windows along the genome and counted the number of 200-bp windows where the GC content was >60% or <20%.

**Whole-genome resequencing**. A total of 15 *E. sinensis* individuals were used for whole-genome resequencing (Supplementary Table 13). DNA was extracted from muscle tissue using the phenol/chloroform extraction method[93] and genomes were sequenced on the Illumina Hiseq X Ten System. Paired-End (PE150) reads from each crab were aligned to the reference genome using Burrows-Wheeler Aligner (BWA) v0.7.12-r1039[94]. SNPs and InDels were called using a Bayesian approach as implemented in the package SAMtools[95]. A total of 27,053,247 bi-alleles SNPs with a missing rate ≤0.2 and MAF (minor allele frequency) ≥5% were used for subsequent analyses. The demographic history was inferred with a pairwise sequentially Markovian coalescence (PSMC)[96] model.

**Gene prediction and annotation**. Three methods, homolog-based, de novo and transcriptome-based predictions, were used for gene prediction for the *E. sinensis* genome. Homologous sequence search was performed by comparing the protein sequences of ten species against the repeat-masked *E. sinensis* genome, using TBLASTN v2.2.26 with *E*-value ≤ 1e−5. The corresponding homologous genome sequences were then aligned with the matching proteins using GeneWise v2.4.1[97] to extract accurate exon–intron information. Four ab initio prediction software programs, Augustus v3.0.2[98], GENSCAN v1.0[99], GlimmerHMM v3.0.4[100], and SNAP v2006-07-28[101], were employed for de novo gene prediction. Results derived from the homology-based and ab initio prediction were integrated to generate a consensus gene set using GLEAN v1.0.1[102] with default parameters. Finally, the assembly unigenes from RNA-seq reads were mapped to the assembly using BLAT v36. Cufflinks v2.2.1[103] was then used to combine the mapping results and to predict transcript structures. To obtain gene function annotations, the predicted protein sequences of *E. sinensis* were aligned to public databases, including NCBI nr, NCBI nt, COG, GO (Gene Ontology)[104], KEGG (Kyoto Encyclopedia of Genes and Genomes)[105], InterPro[106], Swiss-Prot and TrEMBL[107], to predict the gene function.

**Gene family and evolutionary analyses**. The OrthoMCL pipeline v1.02[108] was used to define gene families for 15 genomes (*Eriocheir sinensis, Portunus trituberculatus, Procambarus virginalis, Litopenaeus vannamei, Armadillidium vulgare, Hyalella azteca, Parhyale hawaiensis, Tigriopus californicus, Eurytemora affinis, Eulimnadia texana, Daphnia pulex, Tribolium castaneum, Bombyx mori,*

*Drosophila melanogaster,* and *Tetranychus urticae*). The orthologous genes were aligned via all-against-all BLASTP and clustered with the MCL algorithm. For phylogenetic analysis, single-copy gene families from nine arthropods and *P. yessoensis* were aligned using MUSCLE v3.7[109], and a phylogenetic tree was constructed using PhyML v3.0[110]. The divergence time for *E. sinensis* and other arthropods was estimated using the MCMCTREE program in the PAML package v4.4[111] based on fossil-based calibration times (Supplementary Table 16). Gene family expansion and contraction analysis was performed using the CAFE v1.6[112,113].

**Stress response genes analysis**. To explore the change in RNA expression response to stress, 100 adult female crabs were acclimated for seven days in oxygenated water and 20 °C in individual 120 L buckets. Crabs were fed daily with clams of 5% of total crab weight. Fifty percent of water was changed every day. We examined the crab's stress response to starvation by starving the crabs for 188 h, sampling at 30, 97, and 188 h. To assess crabs' transcriptional responses to desiccation and cold+desiccation stress, we kept 20 crabs without water at 20 °C and 20 crabs in 5 °C chiller, and sampled their hepatopancreas for RNA extraction at 30 h and 97 h, respectively, which was the time when crabs began to show fatality based on our pilot experiment under the same conditions. Crabs cultured in water at 20 °C for 30 h and 97 h was used as control, respectively. Hepatopancreas of the crabs was chosen for transcriptome sequencing by Illumina HiSeq4000 (Lianchuan Biotechnology, Hangzhou, China). Differentially expressed gene (DEG) analysis was carried out using edgeR v3.22.5[114] with three biological replicates, and genes with a fold-change value ≥2 and adjusted *p*-value <0.05 were defined as significant DEGs.

**Osmoregulation-related genes analysis**. Putative osmoregulation-related genes in *E. sinensis* and other arthropods were identified by homology-based searching against the known genes of other animal species retrieved from the NCBI protein database, at an *E*-value threshold of 1e−5. For candidate genes, only those containing complete domains were kept for subsequent analysis. The functional domain analysis was performed using SMART GENOMES (http://smart.embl.de/smart/set_mode.cgi?GENOMIC=1). To validate RNA-seq data and expression profiles obtained from DESeq analysis, megalopae in freshwater and seawater, and the posterior gills of adult crabs in freshwater and seawater were used for qPCR analyses, respectively. The expression levels of key genes were calculated using the $2^{-\Delta\Delta Ct}$ method[115]. The β-actin and 18S rDNA gene were used to normalize the gene expression in megalopae and adults, respectively (Supplementary Data 8). The results were subjected to one way analysis of variance (one way ANOVA) using SPSS 16.0, and the *p*-values less than 0.05 were considered statistically significant.

**Homeobox gene analysis**. Homeobox genes of *E. sinensis* were searched with BLAST in the whole genome assembly of *E. sinensis* using the complete homeobox catalogs of *L. vannamei, P. hawaiensis,* and *D. melanogaster* as queries. Phylogenetic analysis was performed using MEGA5[116] to construct neighbor-joining and maximum likelihood trees. MicroRNAs *miR-993, miR-10,* and *miR-iab-4/8*[50] were identified in *E. sinensis* genome using BLAST with an *E*-value threshold of 1e−5 against the miRNA sequences in *D. melanogaster* downloaded from miRBase[117]. The hit sequences were extracted and aligned with *D. melanogaster* sequences using ClustalW v2.1. The following criteria were used for identifying miR genes as described by Miura et al.[60]: 1) the sequence showed ≥70% sequence identity with *D. melanogaster* sequences at the mature region; 2) free energy of the hairpin structure predicted by the software mfold[118] was ≤ −15 kcal/mol; and 3) the mature sequence was derived from one arm of the hairpin structure. The heat map of Hox gene expression was drawn using Omicshare CloudTools (https://www.omicshare.com/tools/Home/Soft/heatmap). In order to investigate the molecular mechanism of brachyurization metamorphosis, we collected the transcriptome data from early developmental stages (N1, Z1, M1, M2, and P1 stage) of shrimp *L. vannamei*[119], the metamorphosis stages of phyllosoma (20d, 25d), puerulus (clear, H-phase stage) and J1 larva of lobster *S. verreauxi*[120], and the OZs, Z1, EM, LM, and J1 stages of crab *E. sinensis*. The expressions of Hox genes (*Ubx, Abd-A,* and *Abd-B*) involved in posterior thoracic and abdominal development were compared and analysed based on the transcriptome data from the cephalothorax and abdomen of the last megalopa and juvenile instar stages (Novogene Bioinformatics Technology, Beijing, China). The expression of miRNAs was determined by microRNA first-strand synthesis and miRNA quantitation kits (Takara Bio, USA). The primers used for qPCR were listed in Supplementary Table 20.

**Fluorescence in situ hybridization (FISH) of *Abd-A***. Half of the pleons of megalopaes and the first juvenile crabs were fixed in a solution of 4% paraformaldehyde in 0.1 M PBS at 4 °C for overnight. Specific mRNA probes were designed from the sequence of *Abd-A* using Primer 5.0. The probe conjugated to fluorescein isothiocyanate (FITC) was synthesized by Generay Biotechnology (Shanghai, China). The probe sequences conjugated with red fluorophores for *Abd-A* and with green fluorophores for GFP as control. Samples were rinsed for three times at 5 min intervals at room temperature in PBS, followed by 0.3% Triton-X 100 (PBS-T) for 10 min. The samples were digested with protease K (10 μg/mL) for 40 min at 37 °C. After re-fixed by 4% PFA, the samples were hybridized overnight

at 57 °C with probe (300 nM), and then washed with 50% formamide deionized diluted to different concentrations of SSCT (0.1% Tween-20) solution (2× SSCT and 0.2× SSCT). Subsequently, samples were rinsed for five times at 15 min intervals in PBST buffer. Before visualization, samples were incubated in 4′,6-diamidino-2-phenylindole (DAPI, Invitrogen) buffer for 4 min to make the cell nuclei were labeled. Finally, the samples were imaged using a ZEISS LSM880 laser scanning confocal microscope.

**AG network analysis.** Co-expression gene networks were constructed with WGCNA[121] using 29 transcriptome datasets, including at different life history stages (Fe, Cs, Bs, Gs, Hs, Z1, Z5, EM, LM, and J1), in different tissues (eyestalk, gill, hepatopancreas, muscle, middle segment of vas deferens, gonad, testis, and AG) and of treatment of AG (KDC_A, KDT_A, before and after eyestalk ablation EAC_A and EAT_A, sequenced by Novogene Bioinformatics Technology, Beijing, China). The hubness of a gene in a given module was measured by its connection strength with other genes in the module, and was determined by intramodular connectivity (Kwithin)[121]. To identify the AG-related module, over-representation analysis of the AG-related genes (i.e., the differentially expressed genes in the AG relative to adult tissues) was performed for each module using a hypergeometric test with $p$ values adjusted with the Benjamini–Hochberg method[122] for multiple-test correction. We conducted an integrated analysis of miRNA and mRNA expression profiles in AG at synthesis (SY) and secretion (SE) phases to gain a deeper understanding of AG in *E. sinensis*. The sex of larva was determined by two female-specific DNA markers SM_F1/SM_R1 and SM_F2/SM_R2 (Supplementary Table 21). The expression pattern of *IAG* and *iDMY* in sex distinguished larva was detected by qPCR. To test the effect of *iDMY* on *IAG* expression, we examined the expression pattern of *iDMY* and *IAG* in the AG after injection of dsiDMY (see legend of Supplementary Fig. 32). Besides, differentially expressed genes after eyestalk ablation and differentially expressed proteins (sequenced by PTM Bio, Hangzhou, China) involved in IAG secretion were used to predict the synthesis and secretion pathways of IAG.

**Reporting summary.** Further information on research design is available in the Nature Research Reporting Summary linked to this article.

## Data availability

The *Eriocheir sinensis* genome data have been deposited at NCBI under the accession code CL100111224_L02. 10X Genomics data were deposited at the NCBI under the BioProject number PRJNA238496. The genomic Hi-C sequencing data were deposited in the Sequence Read Archive (SRA) database at SRR10802271. RNA-Seq data used for annotation and biological analyses include the following: NCBI SRA SRR2180019-SRR2180020 (https://trace.ncbi.nlm.nih.gov/Traces/sra/?study=SRP062750), SRR770582, SRR769751, SRR1199039, SRR1199058, SRR1205971, SRR1199228, SRR2170964, SRR2170970, SRR10058623-SRR10058634 (https://trace.ncbi.nlm.nih.gov/Traces/sra/?study=SRP220350), SRR10083958-SRR10083963 (https://trace.ncbi.nlm.nih.gov/Traces/sra/?study=SRP220979), SRR10276365-SRR10276369 (https://trace.ncbi.nlm.nih.gov/Traces/sra/?study=SRP225577), SRR10276537-SRR10276548 (https://trace.ncbi.nlm.nih.gov/Traces/sra/?study=SRP225587), SRR13644341-SRR13644350 (https://www.ncbi.nlm.nih.gov/bioproject/?term=PRJNA699917), SRR13664056-SRR13664067 (https://www.ncbi.nlm.nih.gov/bioproject/?term=PRJNA700787) and PRJNA700687. The proteomic data have been deposited to the ProteomeXchange Consortium via the PRIDE partner repository with the dataset identifier PXD024496. The *E. sinensis* genome sequences are also be available at the genome website (http://www.genedatabase.cn/esi_genome.html).

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

## Acknowledgements

This work was supported by grants from the National Key R&D Program of China (2018YFD0900303), the National Natural Science Foundation of China (32072964), the Ten Thousand Talents Program, the Scientific and Technological Innovation Project of Qingdao National Laboratory for Marine Science and Technology (2015ASKJ02), Collaborative Research Fund from the Research Grants Council, Hong Kong Special Administrative Region, China (C4042-14G), and the Ministry of Science and Technology, Taiwan and the Center of Excellence for the Oceans, National Taiwan Ocean University.

## Author contributions

Z.C., F.L., and K.H.C. conceived and designed the study. Z.C. and K.H.C. coordinated the overall project and supervised various parts of the study with F.L. and J.X. Y.L., J.Y., X.Z., CW.S. and C.B. carried out transcriptome analysis and database management. S.S., D.Z., G.F. and Q.C. directed sequencing data generation and genome assembly. D.Z. Q.C., and S.H. performed gene annotation, repetitive elements, and genome evolution. S.S., CC.S. and G.F. performed the Hi-C analysis. T.-Y.C performed morphological identification and drew the life cycle of crab. Y.L., CW.S., Y.Y., H.L., and J.D. performed the laboratory works. Y.L., J.Y., X.Z., and K.Y.M. wrote the manuscript. T.V. advised on development-related content. Q.P.F. and G.G.S. provided access to the eastern spiny lobster transcriptome. J.Q., K.Y.M., M.H., T.V., and K.H.C. critically revised the manuscript. All authors read and approved the final manuscript.

## Competing interests

The authors declare no competing interests.
