## [Peer Review File · Nature Communications]

Editorial note: Neither of the original reviewers were available in the second round of review, so reviewer 3 was brought in to check the revised version.

Reviewers' comments:

Reviewer #1 (Remarks to the Author):

Review for Nature Communications

Title: The Chinese mitten crab genome provides insights into brachyurization and adaptive plasticity

Authors: Cui et al.

Summary: This is a well-written and interesting paper announcing the sequencing of a genome of the Chinese mitten crab. A draft genome was published in 2016. So, in a way, this is a follow-up paper on that genome. This genome, however, has a lot more going for it. It is much more complete. The authors have used state-of-the-art technology for sequencing their genome and for doing assembly and annotations. They use a powerful combination of short-read and long-read sequencing coupled with Hi-C for outstanding genomic resolution.

However, there is not a good comparison of the genome assembly statistics. No N50/N90 values are reported nor compared to other genome assembly efforts on this species or related species. The authors then, in my view, attempt to do way too much with way too little. They conduct an 'evolutionary analysis' of Pancrustacea with a sample size of 13 taxa to represent over 80 orders within the Pancrustacea plus outgroups. I don't think so. They are limited because they also want to perform some comparative genomic analyses and these are the genomes that are available (apparently). That is fine and the comparative genomics is interesting, but leave out the Pancrustacea phylogenetic 'insights' and divergence time estimates as these are meaningless given the sampling. Stick to the genomics. Similarly, the 'population genomics' section is tangential and poorly focused. The authors claim a total of 37 'mitten crabs' from across 5 species for a population genetic study. They provide no information on where these were sampled, how this sampling relates to the different species distributions, and the sampling is simply poor for a population genetic analysis (which would typically include hundreds of individuals per species throughout the distributional range of the species).

I think the genomics is very interesting as is the comparative genomics and inferences relative to development (especially brachyurization) and osmoregulation. This is plenty. The manuscript, in its present form, simply has too much going on, much of which is tangential to the genomics story and not well done. I recommend the authors refocus the paper on the genomics. Provide more information about the contig and scaffold assembly (statistics), do the comparative genomics, but the 'phylogenetics' and 'population genomics'. This will help make the paper more digestible and more helpful to the research community. Below I detail a number of areas where I think the manuscript can be greatly improved.

Genome focus

This paper, first and foremost, is a genome report. That is the major result being communicated and the paper should, therefore, focus on this result. The authors don't even seem to report a genome size based on their assembly. This should be the first table of the results section: # sequences, #bases, N50, N90, N95 for contigs and scaffolds. Then they suggest they have a chromosome level assembly, but I can't even tell how many chromosomes there are in this crab. It would be great to see the first figure of the results showing coverage across chromosome with some SNPs and/or key genes (e.g., for osmoregulation) highlighted. It would be even better if the authors developed a resource that could be used in an online genome browser. I feel the authors were so anxious to get on to the various other things in the manuscript, they did not do justice to the main result of the work – the genome.

Comparative genomics

The authors compare the new genome with 'arthropods' throughout the paper. But often the comparisons are not well articulated. Which other 'arthropods' and why are these reasonable comparisons? It seems the authors are simply taking whatever genomes they can get their hands on and doing comparisons. For example, for osmoregulation, are there other species who have genomes available that have part of their life cycle in marine and part in freshwater? Perhaps this would be the most reasonable comparison for osmoregulatory genes. For brachyurization, then perhaps a sister, non-brachuran, taxon? For High Fertility and stress tolerance, perhaps a non-invasive compared to other invasives. The generic 'compared with other arthropods' (line379) is not at all well justified and I'm not sure the genes identified are therefore meaningful. Perhaps this is just the expected amount of gene family differences across 500 million years of evolution that the authors are comparing?

Drop the population genetics and phylogenetics

As discussed in the summary, the sampling strategies are all wrong for both population genetic and phylogenetic studies (as well as demographic history). The paper would be much clearer if the authors focused on the genomic results. I appreciate the SNP analysis of variation across the '37 populations', but even there, I cannot tell how these 37 'populations' are distributed across the 5 target species. Maybe you have more variation in the target species simply because you have more samples of that species? The reader cannot tell from the paper. If Figure 4 is telling the sampling story, then it looks like one species (*E. hepuensis*) was sampled from one location whereas the target species was sampled around the globe. At any rate, the sampling schemes for the population genetics and the phylogenetics are not well justified.

Minor Issues

Abstract, line 55 'genomic database' – not really a database, more of a 'genomic resource'

Line 111 'quality and integrity of the assembly' please provide quality scores including contig and assembly N50, N90 and compare to other genome assemblies.

Line 146 '60 single-copy orthologous genes', which genes? How do you know they are orthologos?

Figure 3, subfigure 'd' should be 'b'

Line 471 – is there a voucher specimen from the inbred line preserved somewhere as a voucher for the genome? There should be.

Line 569 – why 'one way' analysis of variance? It seems like you should be looking at a two-way for both over expressed and under expressed genes. Also, you should correct for the many, many comparisons going on in such an analysis. You no doubt have a ton of false positives using this statistical approach.

Reviewer #2 (Remarks to the Author):

The manuscript presents one of the most complete genome assemblies for a decapod crustacean so far attempted.

This is a worthy task as the Decapoda is a highly speciose and diverse group with many species being of economic and ecological importance. Further, genomic resources for the decapods are scant, and the sequencing and assembly is challenging as many species have large repetitive genomes, although this species has the smallest of all the species so far sequenced (= ~1.5 Gbp).

Based on my knowledge of genomic and transcriptomic sequencing and assembly and annotations studies, the methodologies and results appear sound technically.

The authors then use this resource for a number of comparative studies, which include: ¹_{SEP}

1. developmental and genetic basis of brachyurization (attainment of the crab-like form) focusing on Hox genes
2. basis of "adaptive plasticity" in the species, defined principally as it related to osmoregulation and fertility
3. Demographic, population and inter-specific genomic variation, using genome-resequencing of 37 mitten crabs from a range of populations and species.

In general, while some of these individual sub-studies and comparisons presented are of potential interest, none are conducted in particular depth, so this collection of smaller comparative studies lack coherence and a focus, which therefore limits the overall value of this manuscript. In addition, the conclusion to many of these isolated studies are vague and lack definitive outcomes or represent "overreach" in terms of cause and effect.

Example are below with words raising doubt or uncertainty or unjustified being highlighted:

Lines 237-240: In summary, the split Hox clusters and miR-iab-4/8 duplication and segment-polarity genes in the genome **might** tighten the regulation of the posterior Hox genes leading to the degeneration of the crab's tail during the brachyurization metamorphosis and the distinctive body configuration of true crabs.

Lines 290-292: **implying** that F-ATPase may be more important than the two other ATPases in the osmoregulatory functioning of the *E. sinensis* gill. ^[SEP]

Lines 383-386: These species-specific and expanded gene families are **probably** important to the *E. sinensis* lineage-specific environmental adaptations, enabling this particular mitten crab to occupy more diverse niches through transitions from pelagic oceanic phases to benthic brackish to freshwater environments.

Lines 394-396: Thus, *E. sinensis* **may** have developed a specific genetic architecture to allow for adaptation to a wider array of ecological niches, explaining its successful invasive nature.

Lines 408 -410: Thus, the faster LD decay in *E. sinensis* **might** be due to its larger effective population size and high fertility, as indicated by its ^[SEP]demographic history. ^[SEP]

Lines 417-420: Two KEGG pathways, namely the GnRH signaling pathway and oocyte meiosis, which play central roles in the ^[SEP]regulation of gonadal development, were over-represented (P<0.05, Supplementary ^[SEP]Table 28), **supporting** the high fertility of *E. sinensis*. (Reviewer: There are many factors - genomic and non-genomic, that will influence fertility - or fecundity).

Lines 437-438: These genetic diversities **may** help *E. sinensis* to adapt to environmental stresses.

^[SEP]Some general concerns that the authors could address to improve the manuscript are:

1. Discuss in more detail and depth the nature and history of the term "brachyurization" which is often used interchangeably with "carcinization". The latter term coined by Borradaile (1916) has arguably a different meaning as "... one of the many attempts by Nature to evolve a crab" (Patsy A. McLaughlin & Rafael Lemaitre Carcinization in the Anomura - fact or fiction? I. Evidence from adult morphology. Contributions to Zoology, 67 (2) 79-123 (1997)). As a consequence it has been argued that a crab-like body form has independently evolved several times (Cunningham et al., 1992: Evolution of king crabs from hermit crab ancestors. Nature 355, 539). The genomic basis of carcinization in different decapod lineages would be an extremely interesting study.
2. The use of the term "adaptive plasticity" need to be better defined. It could refer to environmental flexibility in terms of the animal's ability to physiological cope/exploit diverse habitats or refer to longer terms genetic or evolutionary based adaption. In this context neither

osmoregulation or fertility or population/interspecific genomics are referred to in the Introduction even though they make up significant portions of the reported results.

3. The Introduction lacks an account of how this work fits into the current literature on decapod genomics. On NCBI I note there are genome assemblies for 10 other decapod species from a diversity of Infraorders, not including the recently published *Macrobrachium rosenbergii* genome (Levy et al Scientific Reports volume 9, Article number: 12408 (2019)). Most of these are not discussed or even acknowledged in this manuscript.

4. In short, the Introduction needs to be expanded to provide a more informative context for the study, with more clearly defined aims and objectives and therefore outcomes.

5. Many comparative studies are completed and presented using various combinations of crustacean or Arthropod species. The choice of species used (or not used) needs to be better justified (see point 3 above).

6. STRUCTURE Analysis is used at the population level and is not designed for inter-specific comparisons

7. Overall structure and format of the manuscript: "Introduction" and "Discussion" content are mixed into a long results section and the "Discussion" as written, is little more than a general conclusion.

My recommendation is that the authors sharpen the focus of the paper in terms of their primary aims, place their work firmly within the context of current decapod genome sequencing efforts, and highlight novel conclusions that are unambiguously supported from their data and corresponding analyses.

Reviewer #1 (Remarks to the Author):

Summary: This is a well-written and interesting paper announcing the sequencing of a genome of the Chinese mitten crab. A draft genome was published in 2016. So, in a way, this is a follow-up paper on that genome. This genome, however, has a lot more going for it. It is much more complete. The authors have used state-of-the-art technology for sequencing their genome and for doing assembly and annotations. They use a powerful combination of short-read and long-read sequencing coupled with Hi-C for outstanding genomic resolution.

However, there is not a good comparison of the genome assembly statistics. No N50/N90 values are reported nor compared to other genome assembly efforts on this species or related species.

We have added N50 and N90 values of *E. sinensis* genome and compared these data with other crustacean genomes, including the two published genomes of the same species. Please refer to lines 101-105, 131-133 and the new Supplementary Tables 2 and 3 in the revised MS.

The authors then, in my view, attempt to do way too much with way too little. They conduct an 'evolutionary analysis' of Pancrustacea with a sample size of 13 taxa to represent over 80 orders within the Pancrustacea plus outgroups. I don't think so. They are limited because they also want to perform some comparative genomic analyses and these are the genomes that are available (apparently). That is fine and the comparative genomics is interesting, but leave out the Pancrustacea phylogenetic 'insights' and divergence time estimates as these are meaningless given the sampling. Stick to the genomics.

Thank you for your comments. We now focus the evolutionary analysis to Crustacea instead of Pancrustacea, and include all the good-quality genomes available from the former group. Six genomes are added for comparative genomics, including two Decapoda species *Portunus trituberculatus* and *Procambarus virginalis*, an Amphipoda species *Hyaella Azteca*, an Isopoda *Armadillidium vulgare* and two Copepoda species *Eurytemora affinis* and *Tigriopus californicus*. Accordingly, the descriptions on the estimated divergence time are revised. Please refer to lines 191-196 in the revised MS.

Similarly, the 'population genomics' section is tangential and poorly focused. The authors claim a total of 37 'mitten crabs' from across 5 species for a population genetic study. They provide no information on where these were sampled, how this sampling relates to the different species distributions, and the sampling is simply poor for a population genetic analysis (which would typically include hundreds of individuals per species throughout the distributional range of the species).

We have deleted this section, and added the sampling location of the resequenced individuals of the Chinese mitten crab. In fact, the sampling locations cover the geographical range of the species. Please refer to lines 155-156 in the revised MS.

I think the genomics is very interesting as is the comparative genomics and inferences relative to development (especially brachyurization) and osmoregulation. This is plenty. The manuscript, in its present form, simply has too much going on, much of which is tangential to the genomics story and not well done. I recommend the authors refocus the paper on the genomics. Provide more information about the contig and scaffold assembly (statistics), do the comparative genomics, but the ‘phylogenetics’ and ‘population genomics’. This will help make the paper more digestible and more helpful to the research community. Below I detail a number of areas where I think the manuscript can be greatly improved.

Thank you for your suggestion. We have added N50 and N90 values of *E. sinensis* genome and compared these data with other crustacean genomes. Also we have provided more information about comparative genomics. Please refer to lines 102-105, 131-133, 202-238 and the new Supplementary Tables 2 and 3. As you suggested, we delete some parts on phylogenetics and most parts on population genomics in the revised MS.

Genome focus

This paper, first and foremost, is a genome report. That is the major result being communicated and the paper should, therefore, focus on this result. The authors don't even seem to report a genome size based on their assembly. This should be the first table of the results section: # sequences, #bases, N50, N90, N95 for contigs and scaffolds. Then they suggest they have a chromosome level assembly, but I can't even tell how many chromosomes there are in this crab. It would be great to see the first figure of the results showing coverage across chromosome with some SNPs and/or key genes (e.g., for osmoregulation) highlighted. It would be even better if the authors developed a resource that could be used in an online genome browser. I feel the authors were so anxious to get on to the various other things in the manuscript, they did not do justice to the main result of the work – the genome.

We have added the corresponding results and tables; see lines 102-105, 131-133 and the new Supplementary Tables 2 and 3 in revised MS. The chromosomes no. of *E. sinensis* has been added in lines 119-121. We have shown the SNP density of resequencing individuals with polymorphism hotspot regions colored in red in Fig. 1b5. We have also uploaded the *E. sinensis* genome on the genome browser (http://www.genedatabase.cn/esi_genome.html) as an online genome browser.

Comparative genomics

The authors compare the new genome with ‘arthropods’ throughout the paper. But often the comparisons are not well articulated. Which other ‘arthropods’ and why are

these reasonable comparisons? It seems the authors are simply taking whatever genomes they can get their hands on and doing comparisons.

For example, for osmoregulation, are there other species who have genomes available that have part of their life cycle in marine and part in freshwater? Perhaps this would be the most reasonable comparison for osmoregulatory genes.

Thank you for your suggestion. For comparative analysis of osmoregulation genes, based on the genome data of two decapods (*E. sinensis* and *L. vannamei*), we added two more decapod species (*Portunus trituberculatus* and *Procambarus virginalis*) for the comparison and performed the whole comparative analyses again. Please refer to lines 263-264 and revised Fig. 3b. *P. trituberculatus* is also a crab whose life cycle was significantly different from that of *E. sinensis* as the former spent its whole life in seawater. Besides, desalination play important role in adult sexual maturity and juvenile development in *E. sinensis*, which was also different from *P. trituberculatus*. Like *E. sinensis*, the marbled crayfish *P. virginalis* also inhabiting freshwater, but they spent their whole life in freshwater.

For the new comparative analysis, the results also supported that F-ATPase was a significantly expanded gene family in *E. sinensis*, including the comparison with the other crab *P. trituberculatus*. We also performed Ka/Ks analysis on the orthologous genes of two crabs to identify genes under positive selection (lines 274-278 and 288-291). While none of genes of NKA, V-type and P-type ATPases were positively selected, two genes encoding F-ATPase have $\omega > 1$. The result indicates the two genes may underwent rapid evolution. Thus, it is reasonable to consider that the co-expansion of subunits, co-expression during development, positive selection, and high activity in the posterior gills of F-ATPase identified it as a gene family crucially related to osmoregulation.

For brachyurization, then perhaps a sister, non-brachyuran, taxon?

For brachyurization, as there are very few crustaceans with reference genomes and extensive transcriptome resources, we used the Pacific white shrimp (*Litopenaeus vannamei*, for both genome and transcriptome) and eastern spiny lobster (*Sagmariasus verreauxi*, for transcriptome only) as non-brachyuran species for comparison. Please refer to Fig. 4a and 4c. Actually, we wanted to add the marbled crayfish (*Procambarus virginalis*) as another non-brachyuran species. However, due to the incomplete genome and transcriptome, we cannot identify the ten complete Hox genes in *P. virginalis* for comparison with *E. sinensis*. As for lobster (*S. verreauxi*), although high quality transcriptome resources are available, it lacks genome information, so that we cannot include the information in Fig. 4a.

For High Fertility and stress tolerance, perhaps a non-invasive compared to other invasives. The generic 'compared with other arthropods' (line379) is not at all well justified and I'm not sure the genes identified are therefore meaningful. Perhaps this is just the expected amount of gene family differences across 500 million years of

evolution that the authors are comparing?

We have rewritten the section on gene family expansion, toning down our emphasis on the species-specific fertility, stress tolerance and invasiveness of mitten crab. We have also added RNA-seq results of stress experiment to provide more support to our findings.

Drop the population genetics and phylogenetics

As discussed in the summary, the sampling strategies are all wrong for both population genetic and phylogenetic studies (as well as demographic history). The paper would be much clearer if the authors focused on the genomic results. I appreciate the SNP analysis of variation across the '37 populations', but even there, I cannot tell how these 37 'populations' are distributed across the 5 target species. Maybe you have more variation in the target species simply because you have more samples of that species? The reader cannot tell from the paper. If Figure 4 is telling the sampling story, then it looks like one species (*E. hepuensis*) was sampled from one location whereas the target species was sampled around the globe. At any rate, the sampling schemes for the population genetics and the phylogenetics are not well justified.

Thank you for your comments. We have deleted this section.

Minor Issues

Abstract, line 55 'genomic database' – not really a database, more of a 'genomic resource'

Done. Please refer to line 56.

Line 111 'quality and integrity of the assembly' please provide quality scores including contig and assembly N50, N90 and compare to other genome assemblies.

Done. We have added N50/N90 values of *E. sinensis* genome and compared these data with other crustacean genomes. Please refer to lines 101-105, 131-133 and the new Supplementary Tables 2 and 3.

Line 146 '60 single-copy orthologous genes', which genes? How do you know they are orthologous?

We have added the list of single-copy orthologous genes in Supplementary Table 17. The orthologous genes were aligned via all-against-all BLASTP (v2.2.21) and clustered with the MCL algorithm. The method is routine in evolutionary analysis, and has been used in many genomic papers (e.g., Zhang et al., 2019; Li et al, 2017). Zhang X, Yuan J, Sun Y, Li S, Gao Y, Yu Y, et al., Penaeid shrimp genome provides insights into benthic adaptation and frequent molting. Nat Commun. 2019 Jan

21;10(1):356. doi: 10.1038/s41467-018-08197-4.

Li Y, Sun X, Hu X, Xun X, Zhang J, et al., Scallop genome reveals molecular adaptations to semi-sessile life and neurotoxins. *Nat Commun.* 2017 Nov 23;8(1):1721. doi: 10.1038/s41467-017-01927-0.

Figure 3, subfigure 'd' should be 'b'

Done.

Line 471 – is there a voucher specimen from the inbred line preserved somewhere as a voucher for the genome? There should be.

Yes. This specimen obtained from Panjin Guanghe Crab Industry Co., Ltd was kept in the museum of Institute of Oceanology with the voucher number Lh_M_001.

Line 569 – why 'one way' analysis of variance? It seems like you should be looking at a two-way for both over expressed and under expressed genes. Also, you should correct for the many, many comparisons going on in such an analysis. You no doubt have a ton of false positives using this statistical approach.

We used one way ANOVA because there is only one variable in the analysis. This method is widely used in the real-time PCR analysis, such as Sun et al. (2017) and Su et al. (2020). Actually, we used two-tailed test for both over expressed and under expressed genes to determine the significance (*p*-value).

Sun JJ, Lan JF, Zhao XF, Vasta GR, Wang JX. Binding of a C-type lectin's coiled-coil domain to the Domeless receptor directly activates the JAK/STAT pathway in the shrimp immune response to bacterial infection. *PLoS Pathog.* 2017 Sep 20;13(9):e1006626.

Su Y, Liu Y, Gao F, Cui Z. A novel C-type lectin with a YPD motif from *Portunus trituberculatus* (PtCLEc1) mediating pathogen recognition and opsonization. *Developmental Comparative Immunology*, 2020, 106: 103609.

Reviewer #2 (Remarks to the Author):

The manuscript presents one of the most complete genome assemblies for a decapod crustacean so far attempted.

This is a worthy task as the Decapoda is a highly speciose and diverse group with many species being of economic and ecological importance. Further, genomic resources for the decapods are scant, and the sequencing and assembly is challenging as many species have large repetitive genomes, although this species has the smallest of all the species so far sequenced (= ~1.5 Gbp).

Based on my knowledge of genomic and transcriptomic sequencing and assembly and annotations studies, the methodologies and results appear sound technically.

The authors then use this resource for a number of comparative studies, which include:

1. developmental and genetic basis of brachyurization (attainment of the crab-like form) focusing on Hox genes
2. basis of “adaptive plasticity” in the species, defined principally as it related to osmoregulation and fertility
3. Demographic, population and inter-specific genomic variation, using genome-resequencing of 37 mittens crabs from a range of populations and species.

In general, while some of these individual sub-studies and comparisons presented are of potential interest, none are conducted in particular depth, so this collection of smaller comparative studies lack coherence and a focus, which therefore limits the overall value of this manuscript. In addition, the conclusion to many of these isolated studies are vague and lack definitive outcomes or represent “overreach” in terms of cause and effect.

Thank you for your comments. We have revised the corresponding sections accordingly, which are discussed in detail below. Briefly, in the revised MS, we have deleted the section on demographic, population and inter-specific genomic variation, and incorporated an in-depth study on sexual development.

Example are below with words raising doubt or uncertainty or unjustified being highlighted:

Lines 237-240: In summary, the split Hox clusters and miR-iab-4/8 duplication and segment-polarity genes in the genome **might** tighten the regulation of the posterior Hox genes leading to the degeneration of the crab’s tail during the brachyurization metamorphosis and the distinctive body configuration of true crabs.

Thanks for your suggestion, we have revised the relevant parts with new experimental findings and adjust the wording accordingly. Overall, we proposed a rather substantiated model for brachyuraization for future, more in-depth research.

Lines 290-292: **implying** that F-ATPase may be more important than the two other ATPases in the osmoregulatory functioning of the *E. sinensis* gill.

This sentence has been revised, and “implying” has been replaced by “suggesting”.

Lines 383-386: These species-specific and expanded gene families are **probably**

important to the *E. sinensis* lineage-specific environmental adaptations, enabling this particular mitten crab to occupy more diverse niches through transitions from pelagic oceanic phases to benthic brackish to freshwater environments.

This part has been deleted in the revised MS.

Lines 394-396: Thus, *E. sinensis* **may** have developed a specific genetic architecture to allow for adaptation to a wider array of ecological niches, explaining its successful invasive nature.

This part has been deleted in the revised MS.

Lines 408-410: Thus, the faster LD decay in *E. sinensis* **might** be due to its larger effective population size and high fertility, as indicated by its demographic history.

This part has been deleted in the revised MS.

Lines 417-420: Two KEGG pathways, namely the GnRH signaling pathway and oocyte meiosis, which play central roles in the regulation of gonadal development, were over-represented ($P < 0.05$, Supplementary Table 28), **supporting** the high fertility of *E. sinensis*. (Reviewer: There are many factors - genomic and non-genomic, that will influence fertility - or fecundity).

This part has been deleted in the revised MS.

Lines 437-438: These genetic diversities **may** help *E. sinensis* to adapt to environmental stresses.

This part has been deleted in the revised MS.

Some general concerns that the authors could address to improve the manuscript are:

1. Discuss in more detail and depth the nature and history of the term “brachyurization” which is often used interchangeably with “carcinization”. The latter term coined by Borradaile (1916) has arguably a different meaning as “.. one of the many attempts by Nature to evolve a crab” (Patsy A. McLaughlin & Rafael Lemaitre Carcinization in the Anomura - fact or fiction? I. Evidence from adult morphology. Contributions to Zoology, 67 (2) 79-123 (1997)). As a consequence it has been argued that a crab-like body form has independently evolved several times (Cunningham et al., 1992: Evolution of king crabs from hermit crab ancestors. Nature 355, 539). The genomic basis of carcinization in different decapod lineages would be an extremely interesting study.

We thank the reviewer’s suggestions and bringing up the difference between “brachyurization” and “carcinization”. “Carcinization” or “brachyurization” is the

most important and interesting feature of crabs (Brachyura). Based on the definition of brachyurization of Števcíć (1971), in this MS, we focus on the developmental regulation of “crab-like” transitions. We hope to elucidate the genomic basis of brachyurization in terms of molecular mechanism and evolution, so “brachyurization” is the more appropriate term. We have collected more available genome data in different decapod lineages (such as *Portunus trituberculatus*), and provide further insights into the brachyurization in the revised MS (lines 385-390 and 397-399). These insights also lay a foundation for understanding of “carcinization” in different decapod lineages (see lines 418-431).

2. The use of the term “adaptive plasticity” need to be better defined. It could refer to environmental flexibility in terms of the animal’s ability to physiological cope/exploit diverse habitats or refer to longer terms genetic or evolutionary based adaption. In this context neither osmoregulation or fertility or population/interspecific genomics are referred to in the Introduction even though they make up significant portions of the reported results.

Done. We have clarified its definition in Introduction (lines 78-81).

3. The Introduction lacks an account of how this work fits into the current literature on decapod genomics. On NCBI I note there are genome assemblies for 10 other decapod species from a diversity of Infraorders, not including the recently published *Macrobrachium rosenbergii* genome (Levy et al Scientific Reports volume 9, Article number: 12408 (2019)). Most of these are not discussed or even acknowledged in this manuscript.

Thank you for your suggestion. We have added the background information on decapod genome background in the Introduction section with corresponding references (lines 101-105 and refs. 16-18, 24, 25). We have compared *E. sinensis* genome with genomes of other decapod species: *Portunus trituberculatus*, *Procambarus virginalis* and *Litopenaeus vannamei* in Results section (Supplementary Table 3). And some published decapod genomes, such as *Penaeus monodon*, *Penaeus japonicus*, *Caridina multidentata*, *Cherax destructor* and *Pandalus platyceros*, are omitted as only draft genomes are available. The genome of *Macrobrachium rosenbergii* has not been made publicly available. We contacted the corresponding author who informed us that the genome is the property of a company and it will take some time before the data are made available to the public. Nonetheless, we have also added five recently published crustacean genomes in the comparison, including an Amphipoda species *Hyaella Azteca*, an Isopoda *Armadillidium vulgare* and two Copepoda species *Eurytemora affinis* and *Tigriopus californicus*.

4. In short, the Introduction needs to be expanded to provide a more informative context for the study, with more clearly defined aims and objectives and therefore outcomes.

Done. The introduction has been substantially revised in the revised MS (lines 101-105 and 107-114).

5. Many comparative studies are completed and presented using various combinations of crustacean or Arthropod species. The choice of species used (or not used) needs to be better justified (see point 3 above).

Thank you for your suggestion. We have added six crustaceans in the analysis, including two Decapoda species *Portunus trituberculatus* and *Procambarus virginalis*, an Amphipoda species *Hyaella Azteca*, an Isopoda *Armadillidium vulgare* and two Copepoda species *Eurytemora affinis* and *Tigriopus californicus*. And these are all the crustacean species with good quality genomes for comparative genomics and evolutionary analyses.

6. STRUCTURE Analysis is used at the population level and is not designed for inter-specific comparisons

We have deleted this analysis as the whole resequencing section was deleted.

7. Overall structure and format of the manuscript: “Introduction” and “Discussion” content are mixed into a long results section and the “Discussion” as written, is little more than a general conclusion.

My recommendation is that the authors sharpen the focus of the paper in terms of their primary aims, place their work firmly within the context of current decapod genome sequencing efforts, and highlight novel conclusions that are unambiguously supported from their data and corresponding analyses.

Thanks for your kind suggestion. We have sharpened the focus of the paper on genome characteristics of Chinese mitten crab, osmoregulation, brachyurization and sexual development with reference to the crustacean-specific androgenic hormone. Both Introduction and Discussion sections have been substantially revised following the suggestions from the reviewer.

REVIEWERS' COMMENTS

Reviewer #3 (Remarks to the Author):

This manuscript reports the assembly and annotation of genome sequence of the Chinese mitten crab *Eriocheir sinensis*. This builds on previous drafts and reports analyses of genes involved in stress response, osmoregulation, body plan patterning, and sexual differentiation of this animal.

A combination of short-read, long read and Hi-C were used to provide a very good, chromosome-linked, assembly.

The authors showed that stress-response genes are up regulated under stress conditions and speculate that the expansion of such gene in this species could be adaptive and allow this species to flourish under different environmental conditions.

Similarly, some circumstantial evidence is presented that the ability of this species to live in fresh or sea water is associated with the expansion of the F-ATPase gene family.

The authors report that this species contains 10 Hox genes that are potentially split into two clusters. Analysis of the expression of these genes led the authors to speculate that the lower levels of expression of posterior Hox genes is associated with the abdominal regeneration of crabs.

Finally, a co-expression network that may represent the regulation of the androgenic gland is presented.

Overall the paper is well written and illustrated, and reports potentially interesting associations between the genomic content and aspects of the biology of this crab (although these insights are mainly based on associations between the expression of genes are particular traits rather than directly tested).

Minor comments

The abstract lacks any background as to why the genome of this animal merits interest, what could be found by studying it and why the main findings of the study are insightful and important. This is developed in the Introduction but it is also need (briefly) in the Abstract for non-specialist readers.

I am surprised to see so many species-specific genes (sup figure 12). This could be because only a few species were looked at but can the authors comment on why there are so many species-specific genes even among the crustaceans analysed?

Lines 66-68 and 302: the final larval stage (megalopa) is defined twice

Line 83: "fitness" is ambiguous – previously criticised for their use of adaptive plasticity – I think they could just remove this term completely? Because they are predominantly referring to osmoregulation

Lines 85-88 and lines 256-258: catadromous species described twice – that being said there are two life cycle diagrams, can they not be made into one figure?

Line 324: (i) and (ii) in figure legend not shown on the life cycle of *E. sinensis* Figure 3a.

Lines 296-297: "F0 subunit c" and then "F0 subunit c (FHA10)" abbreviation after first introduction.

Lines 351-352: Figure 3d – include freshwater (blue) and seawater (orange)

Line 542: "216 gens" correct spelling.

Line 544: transcription factor abbreviation not needed

The information about the first two draft genomes of *E. sinensis* in the N50 comparison in the supplementary appears to be missing?

REVIEWERS' COMMENTS

Reviewer #3 (Remarks to the Author):

This manuscript reports the assembly and annotation of genome sequence of the Chinese mitten crab *Eriocheir sinensis*. This builds on previous drafts and reports analyses of genes involved in stress response, osmoregulation, body plan patterning, and sexual differentiation of this animal.

A combination of short-read, long read and Hi-C were used to provide a very good, chromosome-linked, assembly.

The authors showed that stress-response genes are up regulated under stress conditions and speculate that the expansion of such gene in this species could be adaptive and allow this species to flourish under different environmental conditions.

Similarly, some circumstantial evidence is presented that the ability of this species to live in fresh or sea water is associated with the expansion of the F-ATPase gene family.

The authors report that this species contains 10 Hox genes that are potentially split into two clusters. Analysis of the expression of these genes led the authors to speculate that the lower levels of expression of posterior Hox genes is associated with the abdominal regeneration of crabs.

Finally, a co-expression network that may represent the regulation of the androgenic gland is presented.

Overall the paper is well written and illustrated, and reports potentially interesting associations between the genomic content and aspects of the biology of this crab (although these insights are mainly based on associations between the expression of genes are particular traits rather than directly tested).

Thank you for your comments.

Minor comments

The abstract lacks any background as to why the genome of this animal merits interest, what could be found by studying it and why the main findings of the study are insightful and important. This is developed in the Introduction but it is also need (briefly) in the Abstract for non-specialist readers.

Response: Thank you for your suggestion. We have added the corresponding background. Please see lines 47-49.

I am surprised to see so many species-specific genes (sup figure 12). This could be because only a few species were looked at but can the authors comment on why there are so many species-specific genes even among the crustaceans analysed?

Response: In the comparative genomics analysis, crustaceans have a large number of species-specific genes. Similar results have been also reported in the analysis of penaeid shrimp genome (Ref. 16). We agree with the reviewer that this could be

because too few genomes have been analyzed. This is stated in lines 200-201.

Lines 66-68 and 302: the final larval stage (megalopa) is defined twice.

Response: We have deleted the repeated definition. Please see line 299.

Line 83: “fitness” is ambiguous – previously criticised for their use of adaptive plasticity – I think they could just remove this term completely? Because they are predominantly referring to osmoregulation

Response: We have deleted this term. Please see line 82.

Lines 85-88 and lines 256-258: catadromous species described twice – that being said there are two life cycle diagrams, can they not be made into one figure?

Response: Done. We have combined the two figures into Fig. 1a.

Line 324: (i) and (ii) in figure legend not shown on the life cycle of *E. sinensis* Figure 3a.

Response: Fig. 3a was deleted.

Lines 296-297: “F0 subunit c” and then “F0 subunit c (FHA10)” abbreviation after first introduction.

Response: We have changed the description to be clearer. FHA10, belonging to the members of F0 subunit c, is the gene identified from the *E. sinensis* genome. We have changed the corresponding description. Please see lines 295 and 345.

Lines 351-352: Figure 3d – include freshwater (blue) and seawater (orange)

Response: Done. We have included the notes of freshwater and seawater. Please see lines 344-345.

Line 542: “216 gens” correct spelling.

Response: Done. We have corrected the spelling. Please see line 535.

Line 544: transcription factor abbreviation not needed

Response: Done. We have deleted the abbreviation. Please see line 536.

The information about the first two draft genomes of *E. sinensis* in the N50 comparison in the supplementary appears to be missing?

Response: Done. We have added the information. Please see the revised Supplementary Table 2.